# Apolipoprotein M-bound sphingosine-1-phosphate regulates blood–brain barrier paracellular permeability and transcytosis

Mette Mathiesen Janiurek[1], Rana Soylu-Kucharz[2], Christina Christoffersen[3,4], Krzysztof Kucharz[1†]*, Martin Lauritzen[1,5†]*

[1]Department of Neuroscience, University of Copenhagen, Copenhagen, Denmark; [2]Department of Experimental Medical Science, Lund University, Lund, Sweden; [3]Department of Clinical Biochemistry, Rigshospitalet, Copenhagen, Denmark; [4]Department of Biomedical Sciences, Copenhagen University, Copenhagen, Denmark; [5]Department of Clinical Neurophysiology, Rigshospitalet-Glostrup, Copenhagen, Denmark

**Abstract** The blood-brain barrier (BBB) is formed by the endothelial cells lining cerebral microvessels, but how blood-borne signaling molecules influence permeability is incompletely understood. We here examined how the apolipoprotein M (apoM)-bound sphingosine 1–phosphate (S1P) signaling pathway affects the BBB in different categories of cerebral microvessels using ApoM deficient mice ($Apom^{-/-}$). We used two-photon microscopy to monitor BBB permeability of sodium fluorescein (376 Da), Alexa Fluor (643 Da), and fluorescent albumin (45 kDA). We show that BBB permeability to small molecules increases in $Apom^{-/-}$ mice. Vesicle-mediated transfer of albumin in arterioles increased 3 to 10-fold in $Apom^{-/-}$ mice, whereas transcytosis in capillaries and venules remained unchanged. The S1P receptor 1 agonist SEW2871 rapidly normalized paracellular BBB permeability in $Apom^{-/-}$ mice, and inhibited transcytosis in penetrating arterioles, but not in pial arterioles. Thus, apoM-bound S1P maintains low paracellular BBB permeability in all cerebral microvessels and low levels of vesicle-mediated transport in penetrating arterioles.

*For correspondence:
kucharz@sund.ku.dk (KK);
mlauritz@sund.ku.dk (ML)

†These authors contributed equally to this work

Competing interests: The authors declare that no competing interests exist.

## Introduction

The blood-brain barrier (BBB) allows only certain molecules and cells to pass from the blood circulation to the brain. Molecules cross the BBB via the paracellular route by diffusion between endothelial cells (*Tietz and Engelhardt, 2015*), or by transcellular route, including vesicular transport across the endothelium (*De Bock et al., 2016*). The selectivity of the BBB towards small molecules is maintained by tight and adherens junctions between adjoining endothelial cells (*Tietz and Engelhardt, 2015*), whereas the transport of large molecules is restricted by receptor-mediated interactions (*De Bock et al., 2016*). Under normal conditions, the BBB maintains low permeability due to the structural barrier maintained by junctional complexes and the low rate of transcytosis (*De Bock et al., 2016; Daneman and Prat, 2015; Nico and Ribatti, 2012; Abbott et al., 2010*). In disease states, the loss of BBB function leads to increased permeability, which is viewed mainly as the consequence of disrupted junctional complexes at endothelial cell contact sites (*Davies, 2002; Engelhardt et al., 2014; Salameh et al., 2016*). However, recent evidence suggests equal importance of transcytosis, as a failure in the homeostatic regulation of endothelial trans-cellular transport may aggravate brain pathologies (*Knowland et al., 2014; Habgood et al., 2007; Hashizume and Black, 2002; Krueger et al., 2013*).

Sphingosine-1-phosphate (S1P) is a bioactive sphingolipid synthesized by all mammalian cells, with blood-circulating S1P produced by vascular endothelial cells, platelets, and erythrocytes

(*Thuy et al., 2014*; *Blaho and Hla, 2014*; *Xiong and Hla, 2014*; *Prager et al., 2015*). S1P is a ligand for five membrane-bound G-protein-coupled receptors (S1PR1-5), and the brain endothelium expresses S1PR1-3 (*Prager et al., 2015*). The majority (~70%) of S1P in the blood circulation is transported by apolipoprotein M (apoM), a 26 kDa protein mainly associated with high-density lipoprotein (HDL) (*Xu and Dahlbäck, 1999*), while the remaining S1P fraction is carried primarily by albumin (*Ruiz et al., 2017*; *Christoffersen et al., 2011*).

Depending on the carrier (i.e., apoM or albumin), S1P preferentially activates S1PR1 or S1PR2 and S1PR3, which can exert opposite effects (*Murata et al., 2000*). For example, S1PR1 stimulation inhibits neuroinflammation, whereas S1PR2 and S1PR3 stimulation facilitates pro-inflammatory signaling cascades (*Sanchez et al., 2007*; *Sanchez, 2016*; *Blaho et al., 2015*). The onset and severity of sepsis in humans correlate with reduced apoM-S1P plasma concentrations and loss of BBB integrity (*Davies, 2002*; *Kumaraswamy et al., 2012*). In mice, lack of apoM compromises the endothelial barrier in the lungs and brown adipose tissue (*Christoffersen et al., 2011*; *Christensen et al., 2016*; *Christoffersen et al., 2018*), but its effects on the BBB are largely unexplored. First, apoM-deficient mice are less protected against neuroinflammation (*Blaho et al., 2015*). Second, conditional knock-out of S1PR1 increases the BBB permeability towards small molecules (*Yanagida et al., 2017*), but no data are available on how the BBB properties change with consistent S1PR1 expression, but during reduced apoM levels.

Our objective was to determine the modulatory role of apoM for the BBB. We used two-photon microscopy to image the brains of apoM-deficient (*Apom*^-/-^) mice in vivo in order to characterize changes in BBB permeability in different categories of cerebral microvessels. Lack of apoM increased the BBB permeability to small molecules without affecting the ultrastructural components of junctional complexes and caused substantial increases in adsorptive transcytosis in pial and penetrating arterioles, but not in venules and capillaries. Systemic administration of a selective S1PR1 agonist SEW2871 promptly reversed the increased permeability for the paracellular route, whereas the normal rate of transcytosis was restored only in penetrating arterioles. We suggest that modulation of the apoM signaling axis may be clinically relevant, but heterogeneous responses of distinct vessel types to apoM shortage and S1PR1 agonist treatment should be taken into account in the development of protective strategies in neurological conditions.

## Results

### Deficiency in apoM increases BBB permeability

To assess the BBB with all structural constituents we used two-photon fluorescence imaging in vivo in transgenic apoM-deficient mice (*Apom*^-/-^) mice (*Christoffersen et al., 2011*), which in contrast to other animal models, e.g. *S1pr1*^iECKO^ (*S1pr1* knock-out) mice (*Yanagida et al., 2017*), retain expression and function of S1PR1 (*Christoffersen et al., 2011*). Following preparative surgery (*Figure 1a*), the brain was imaged in living anesthetized animals through an acute craniotomy over the somatosensory cortex (*Figure 1b*).

First, the integrity of the BBB was assessed after an i.v. bolus injection of sodium fluorescein (NaFluo). NaFluo is a small (0.365 kDa), negatively charged hydrophilic molecule, which determines its preferred route across the BBB via paracellular diffusion, and has been shown to be suitable for characterizing the paracellular leak, independently from vesicular transport across the barrier (*Cheng et al., 2010*; *Hawkins and Egleton, 2006*; *Kozler and Pokorný, 2003*).

The fluorescence signal from blood-circulating fluorophore and brain parenchyma was simultaneously measured for 30 min in a series of Z-stacks of relatively large brain volume (750 µm x 750 µm x 114 µm) that contained pial and penetrating microvessels, and capillaries (*Figure 1c*). Each Z-stack underwent dimensionality reduction, that is, maximum intensity projection along the Z (depth) axis. The changes in fluorescence in brain parenchyma were not apparent on raw fluorescence data images (*Figure 1c*), therefore we chose to represent the data as relative increases of fluorescence over the baseline (*Figure 1d–e*).

The morphology of the vasculature delineated by blood-circulating NaFluo did not differ between WT and *Apom*^-/-^ mice (*Figure 1d*). We detected a progressive accumulation of NaFluo in the brain parenchyma in both WT and apoM-deficient mice. However, *Apom*^-/-^ mice exhibited considerably higher levels of fluorophore accumulation in the brain compared to WT mice already at 15 min after

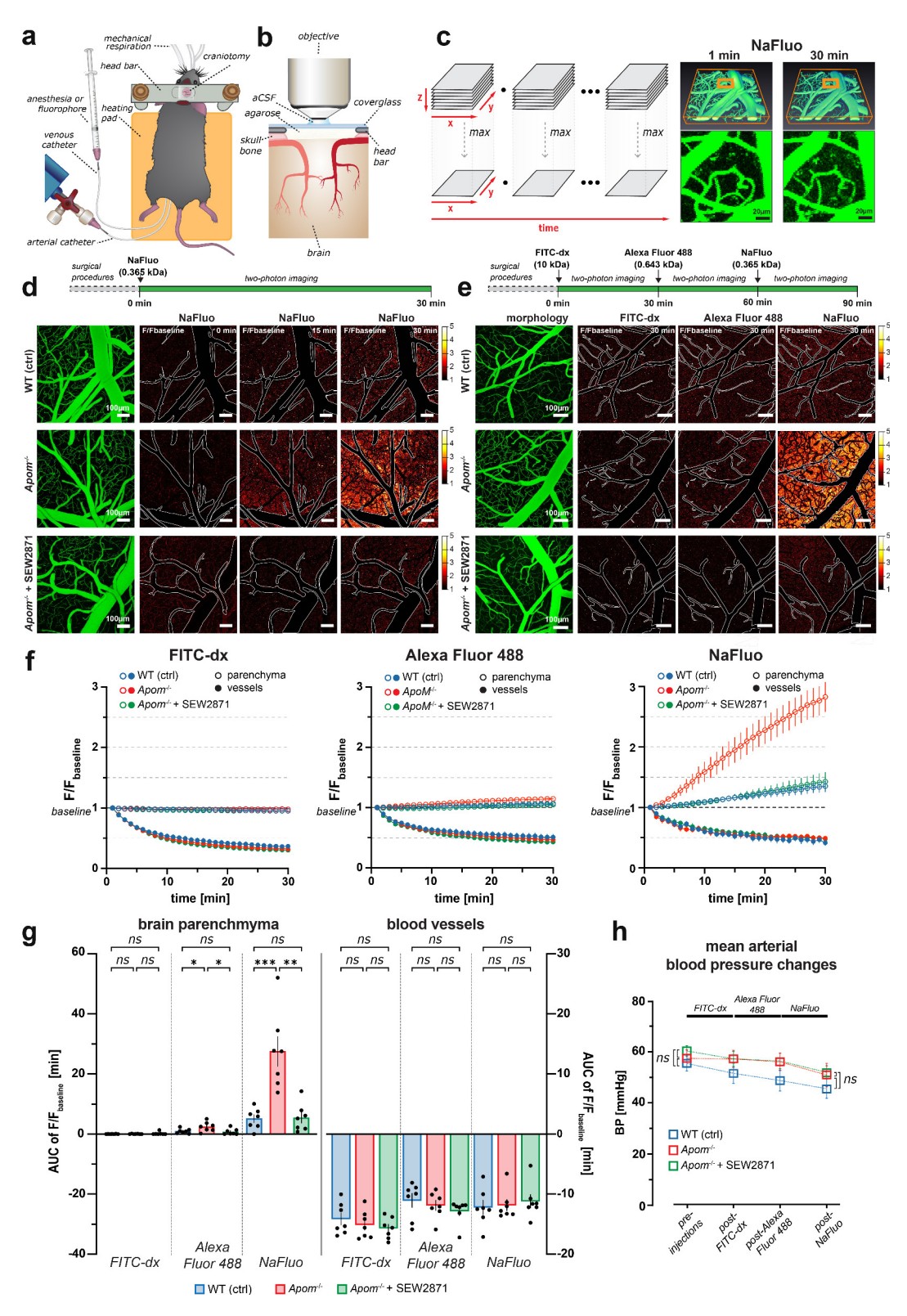

**Figure 1.** Two-photon imaging of paracellular permeability changes in *Apom*⁻/⁻ mice. (a) Schematic of a mouse after preparative microsurgery for two-photon imaging, with (b) craniotomy over the somatosensory cortex. (c) Three-dimensional reconstruction of the brain microvasculature from a single Z-stack after i.v. injection of NaFluo. Each Z-stack corresponds to a single time point and underwent dimensionality reduction by maximum intensity projection along the Z-axis (depth). (d) Relative fluorescence change over the baseline (first projected Z-stack in the recording). Contours delineate the

*Figure 1 continued on next page*

*Figure 1 continued*

main cerebral microvasculature. Compared to WT, the *Apom*⁻ᐟ⁻ animals had increased accumulation of NaFluo in the brain parenchyma over time, which was ameliorated by treatment with S1PR1 agonist SEW2871. (**e**) Bolus injections of fluorophore were separated by 30 min time-lapse imaging intervals. The panels show fluorescent molecule accumulation in the brain parenchyma for all tested fluorophores at 30 min post-injection. (**f**) Changes in fluorescence intensity over time in the brain parenchyma and the blood circulation expressed as a fraction of the baseline. (**g**) Quantitative assessment of the total effect 30 min post-injection by area under the curve (AUC) calculation. Left bar plot: increased paracellular permeability of the BBB in *Apom*⁻ᐟ⁻ mice towards sodium fluorescein (0.365 kDa) and Alexa Fluor488 (0.643 kDa), but not large molecule FITC-dextran (10 kDa). Right bar plot: no differences in the signal decrease of blood-circulating fluorophores among all experimental groups. (**h**) The mean arterial blood pressure during imaging was the same in all experimental groups. All data are presented as mean ± SEM. *p<0.05, **p<0.01, ***p<0.001.
The online version of this article includes the following figure supplement(s) for figure 1:

**Figure supplement 1.** Data analysis: (**a**) Brain parenchyma fluorescence traces from three consecutively injected fluorophores with an exponential curve fit to the FITC-dx trace.

injection, with a clear difference at the 30 min time-point (*Figure 1d*, *Video 1*). As apoM is the main carrier for S1P in the blood stream and S1P bound to apoM primarily activates S1PR1 on the endothelial cells (*Prager et al., 2015*; *Murata et al., 2000*), we tested whether the increase in BBB permeability due to the lack of apoM was a consequence of the S1PR1 hypostimulation. The *Apom*⁻ᐟ⁻ mice were i.p. injected 150 min prior to imaging with a selective S1PR1 agonist, SEW2871. Compared to untreated *Apom*⁻ᐟ⁻, the SEW2871-treated *Apom*⁻ᐟ⁻ mice exhibited decreased permeation of NaFluo into the brain (*Figure 1d*, *Video 1*), suggesting that the S1PR1 agonist rescued the abnormal *Apom*⁻ᐟ⁻ BBB phenotype. This result indicates that the apoM deficiency impairs S1PR1 signaling to the degree that causes increased BBB permeability. We next examined whether the BBB also exhibited increased permeability towards other molecules of different sizes and chemical properties. Each animal in the WT, *Apom*⁻ᐟ⁻, and *Apom*⁻ᐟ⁻ SEW2871-treated group was injected at 30 min intervals with fluorophores of decreasing molecular size, starting with 10 kDa FITC-dextran, followed by 0.643 kDa Alexa Fluor 488 and 0.365 kDa NaFluo (*Figure 1e*). After each injection, we imaged the same brain volume over time, with subsequent dimensionality reduction of the Z-stacks. To compare animal groups, we measured the changes in fluorescence signal over time from ROIs placed in the brain parenchyma and vasculature (n = 7 mice in each group), and accounted for systematic error in measurement of consecutively injected fluorophores (see Methods section, *Figure 1—figure supplement 1*).

For both tracer molecules, Alexa Fluor 488 and NaFluo, and in all animal groups, we detected fluorescence increase over time in the parenchyma, and the effect was persistent without the signal reaching saturation (*Figure 1f*). In contrast, there was no increase in brain fluorescence for FITC-dextran (*Figure 1e–f*). To better characterize the fluorescence increase over time, we calculated the area under the curve (AUC) of the signal traces, that is, the cumulative fluorescence increase for the duration of 30 min after each fluorophore injection (*Figure 1f–g*). We observed no differences in BBB permeability between WT and *Apom*⁻ᐟ⁻ mice for 10 kDa FITC-dextran (WT 0.0086 ± 0.0056 min vs. *Apom*⁻ᐟ⁻ 0.033 ± 0.023 min, p=0.3068), but

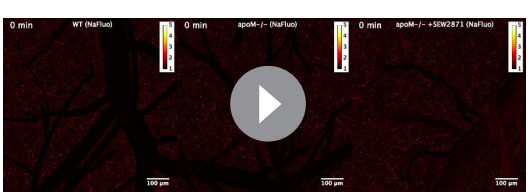

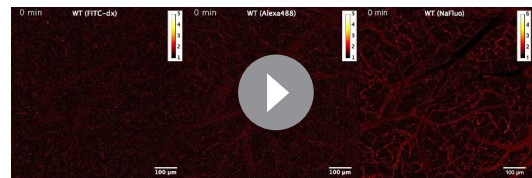

**Video 1.** Lack of apoM leads to a reversible increase in BBB paracellular permeability. Accumulation of NaFluo in brain parenchyma presented as fluorescence increase relative to the baseline. Fluorophore injections occurred 1 min before the first recorded imaging plane.
https://elifesciences.org/articles/49405#video1

**Video 2.** WT mice BBB paracellular permeability towards NaFluo, Alexa Fluor 488, and 10 kDa FITC-dx. Accumulation of fluorophores in brain parenchyma presented as fluorescence increase relative to the baseline. For each time-lapse recording, the respective fluorophore was injected 1 min before the first recorded imaging plane.
https://elifesciences.org/articles/49405#video2

the BBB in *Apom*[-/-] mice was significantly more permeable to small molecules, such as Alexa Fluor 488 (WT 0.90 ± 0.31 min vs. *Apom*[-/-] 2.43 ± 0.61 min, p<0.05) and NaFluo (WT 5.11 ± 1.3 min vs. *Apom*[-/-] 27.5 ± 4.9 min, p<0.001) (*Figure 1g*, *Videos 2–3*).

Treatment of the *Apom*[-/-] mice with SEW2871 markedly reduced the leakage of both Alexa Fluor 488 and NaFluo compared to untreated *Apom*[-/-] mice (0.75 ± 0.38 vs. 2.43 ± 0.61 min, p<0.05; and 5.45 ± 1.6 min vs. 27.5 ± 4.9 min, p<0.01; *Figure 1e–g*, *Videos 3–4*). The reversal of the dysfunctional BBB phenotype occurred relatively fast, and at 150 min post-treatment with SEW2871, the BBB permeability was the same in *Apom*[-/-] and WT mice (p=0.6949 and p=0.8778 for Alexa Fluor 488 and NaFluo, respectively; *Figure 1g*). In addition we performed the same analysis on arbitrary fluorescence units [a.u.], that is non-normalized data (*Figure 1—figure supplement 1e–f*) and the results were in accord with relative fluorescence increases. We chose to report the data as relative changes for the results to be more easily comparable between different imaging setups.

Importantly, the decrease in fluorescence signal in vessels (i.e., clearance of the respective fluorophore from the blood circulation) did not differ between WT, *Apom-/-*and *Apom-/-* SEW2871-treated mice for each respective fluorophore (*Figure 1f–g*). In addition, we found no differences in mean arterial blood pressure between animal groups before (pre-imaging) or after the administration of all fluorophores (post-imaging) (*Figure 1h*). Thus, the increased fluorescence accumulation in the brains of *Apom*[-/-] mice and reduced accumulation of fluorophores in the brains of SEW2871-treated *Apom*[-/-] mice were not caused by different kinetics of a fluorophore clearance from the blood stream or differences in blood pressure.

These results show that apoM shortage increases the BBB permeability towards small molecules (~0.3–0.7 kDa), and that the effect can be reversed by S1PR1 stimulation.

## Deficit in apoM signaling does not alter BBB tight junction ultrastructure

Increases in the BBB permeability towards small molecules may be indicative of defective structural elements that restrict paracellular diffusion across the BBB, e.g. junctional complexes (*Davies, 2002*; *Engelhardt et al., 2014*; *Salameh et al., 2016*). Therefore, we next assessed the ultrastructure of the BBB using transmission electron microscopy (TEM).

We analyzed brain microvessels (<6 μm) cross-sections from WT, *Apom*[-/-], and SEW2871-treated *Apom*[-/-] mice. In contrast to capillaries, large vessels were susceptible to perfusion-fixation artifacts (non-uniform distortion of endothelium neighboring large tissue-devoid areas, i.e. vessel lumen), which rendered pial and penetrating vessels not suitable for TEM quantitative assessment of the BBB ultrastructure. In each vessel, we measured the thickness of the endothelium, basement membrane (both n = 104 vessels; four mice in each group), tight junctional complexes, and the endothelial cleft, the percentage coverage of endothelial cell contact sites by junctional complexes (all n = 164, four mice in each group), and the endothelial cell area (n = 104 vessels, four mice in each group) (*Figure 2*, *Figure 2—figure supplement 1*). The width, length, and coverage of junctional complexes were the same for all three groups of mice (*Figure 2a–b*, *Figure 2—figure supplement 1a*), suggesting that the compromised BBB was not caused by a structural defect of junctions, and that the rescue effect of SEW2871 was not mediated by alteration of the morphology of these structural components. However, we detected an ~ 10% increase in the thickness of the endothelium in S1P-signaling-deficient mice compared to controls (*Apom*[-/-] 0.078±0.0026 μm vs. WT 0.071 ± 0.0025 μm; p<0.05) and the rescue group (*Apom*[-/-]+SEW2871 0.070 ± 0.0025 μm; p<0.05; *Figure 2c*). Notably, the stimulation of S1PR1 in *Apom*[-/-] mice decreased the endothelial cell area (5.80 ± 0.37 vs. 4.24 ± 0.23 μm²; p<0.001), with a strong trend for the cell area in *Apom*[-/-] to be larger than in WT mice (5.80 ± 0.37 vs. 4.78 ± 0.24 μm²; p=0.068; *Figure 2d*). These changes were relatively small,

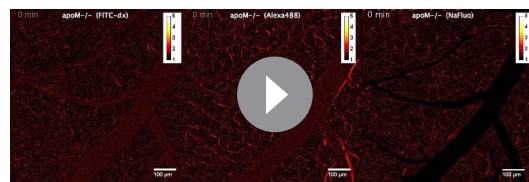

**Video 3.** Increase in the BBB paracellular permeability in *Apom*[-/-] mice towards NaFluo and Alexa Fluor 488, but not 10 kDa FITC-dx. Accumulation of fluorophores in brain parenchyma presented as fluorescence increase relative to the baseline. For each time-lapse recording, the respective fluorophore was injected 1 min before the first recorded imaging plane.
https://elifesciences.org/articles/49405#video3

and they are unlikely to disrupt BBB structural integrity to induce robust permeability increase for small molecules. Altogether, our results suggest that the increased permeability observed in $Apom^{-/-}$ mice is not mediated by ultrastructural changes at the level of the junctional complexes.

## ApoM/S1PR1 signaling deficit increases transcellular transport

Next, we examined whether lack of apoM leads to increased transcytosis in the BBB. The animals were injected with bovine serum albumin conjugated to Alexa 488 (BSA-Alexa 488) into the blood stream to monitor vesicular transport across the endothelium (*Knowland et al., 2014*). Following the injections, the animals were continuously imaged for 120 min in a hyperstack mode with subsequent dimensionality reduction (as in *Figure 1c*). In all animal groups, and as early as 30 min post-injection, we detected BSA-Alexa 488-containing vesicles at the BBB interface (*Figure 3*). Compared to WT animals, $Apom^{-/-}$ mice exhibited increased albumin in endothelial cells, indicating that vesicular transport was increased in the absence of apoM. Furthermore, the treatment of $Apom^{-/-}$ animals with SEW2871 reversed the observed effect, suggesting that the same apoM-dependent mechanism underlies the previously observed increase in permeability for small fluorescent tracers, as well as elevated transcytosis.

We detected no signs of fibrinogen entry into the brain parenchyma; and similarly to S1PR1 conditional KO mice (*Yanagida et al., 2017*), there were no differences in morphology of microglia/macrophages and astrocytes between WT and $Apom^{-/-}$ mice, suggesting lack of gliosis upon exposure to albumin (*Figure 3—figure supplement 1*). However, we detected in the brain parenchyma the presence of cells that sequestered blood-borne albumin (*Figure 3*). To identify the cells, we performed a series of immunohistochemical analysis on PFA-fixated brains of $Apom^{-/-}$ mice to investigate for the colocalization of cell-internalized BSA-Alexa488 with the immunofluorescence signal from ionized calcium binding adaptor molecule 1 (Iba-1, microglia/macrophages marker); transmembrane protein 119 (TMEM119, microglia-specific marker); RM0029-11H3 (monoclonal antibody against macrophage-specific epitope) and glial fibrillary acidic protein (GFAP, astroglia marker). First, we ascertained that the fixation protocol did not quench the signal from BSA-Alexa488 (*Figure 3—figure supplement 2a*). The albumin sequestering cells were identified to be macrophages based on the colocalization of BSA-Alexa488 –positive cells with Iba-1 (microglia/macrophages marker), and further with monoclonal anti-macrophage antibody RM0029-11H3 signal (*Figure 3—figure supplement 2b*). There was no colocalization of the BSA-Alexa488 signal with the microglia-specific marker (TMEM119) and astroglia (GFAP) (*Figure 3—figure supplement 2c*). We found only minor signs (single puncta) of BSA-Alexa488 uptake by other cell types (*Figure 3—figure supplement 2b–c*).

## S1P signaling deficit elevates transcytosis only in arterioles

Next, we assessed whether the increase in transcellular transport caused by the S1PR1-signaling deficiency was the same for all vessel types. The number of vesicles was quantified over time for each individual vessel type: pial arterioles, penetrating arterioles, capillaries, post-capillary venules, ascending venules, and pial venules (n = 5 mice). The data are expressed as the vesicle surface density, expressed as the number of vesicles per square micrometer of vessel surface area (*Figure 4a*). In all experimental groups, the distribution of BSA-Alexa 488-positive vesicles was dependent on vessel type and diameter. At 120 min post-injection, the highest vesicle density in WT mice was observed in capillaries and gradually decreased with vessel diameter, regardless of the vessel type (i.e., arterioles or venules) (*Figure 4a*). In $Apom^{-/-}$ mice, the deficiency in S1PR1 signaling selectively increased transcytosis in pial and penetrating arterioles, but not in venules and capillaries. The effect was most pronounced in penetrating arterioles, where the vascular vesicle density in $Apom^{-/-}$ mice was 10-fold higher than in WT mice ($Apom^{-/-}$ 0.30 ± 0.071 vs. WT 0.038 ± 0.009 vesicles/100 µm²; p<0.01), and three times higher than for pial arterioles ($Apom^{-/-}$ 0.099±0.0023 vs. WT 0.035 ± 0.0011 vesicles/100 µm²; p<0.05; *Figure 4b*). The level of transcytosis did not change in $Apom^{-/-}$ mice in the capillaries, venules, or pial veins (p=0.2306, p=0.3562, and p=0.9546, respectively).

## Penetrating, but not pial arterioles respond to SEW 2871 treatment

S1PR1 stimulation with SEW2871 led to normalization of the transcellular transport of albumin in the penetrating arterioles of $Apom^{-/-}$ mice to the level of WT controls ($Apom^{-/-}$+SEW2871 0.68 ± 0.016

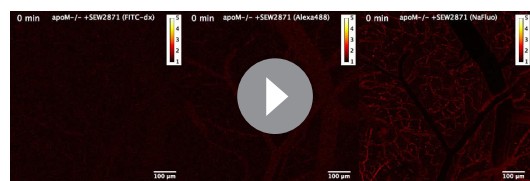

**Video 4.** *Apom*[-/-] SEW2781-treated mice exhibit normal BBB paracellular permeability towards NaFluo, Alexa Fluor 488, and 10 kDa FITC-dx. Accumulation of fluorophores in brain parenchyma presented as fluorescence increase relative to the baseline. For each time-lapse recording, the respective fluorophore was injected 1 min before the first recorded imaging plane. https://elifesciences.org/articles/49405#video4

vs. WT 0.038 ± 0.009 vesicles/100 $\mu m^2$, p=0.1156; *Figure 4b*). In contrast, SEW2871 treatment did not decrease transcytosis in pial arterioles (*Apom*[-/-]+SEW2871 0.84 ± 0.017 vs. *Apom*[-/-] 0.99±0.023 vesicles/100 $\mu m^2$; p=0.4879), and albumin transport was still significantly elevated compared to WT mice (*Apom*[-/-]+SEW2871 0.84 ± 0.017 vs. WT 0.35 ± 0.011 vesicles/100 $\mu m^2$; p<0.05). Lastly, we assessed the kinetics of the vesicular density increase over time with respect to each animal group and vessel type. In all vessel types and animal experimental groups, the increase in BSA-Alexa 488 vesicle surface density was persistent and did not reach saturation at 120 min, indicating that the results were not obscured by overloading the transcellular transport with BSA-Alexa 488 (*Figure 4c*).

These findings show that transcellular transport is upregulated in apoM-deficient mice, and that there is not only heterogeneity in the vasculature susceptibility towards increased permeability, but also towards which vessels can be rescued by stimulation of the S1PR1 signaling pathway.

## Discussion

Understanding the influence of blood-borne signaling molecules on brain endothelial cells is critical for an understanding of the BBB function. We report that lack of apoM increased the BBB permeability for small molecules without changes in size, structure, magnitude, or subcellular localization of tight junction complexes. Furthermore, we report an up to 10–fold increase in adsorptive transcytosis in the endothelial cells of pial and penetrating arterioles, but not in venules and capillaries. Thus, ApoM modulates the flux of small and large molecules to different extents in different vascular segments. Finally, the increased BBB permeability for small molecules and transcytosis-mediated large proteins could be rescued by selective stimulation of the S1PR1. So far, the two-photon assessments of the BBB in vivo are scarce. The presented data is first to show the heterogeneity of the BBB responses to S1PR1 impairment and treatment in different vessel types, thus underlines the necessity of studying the BBB in a living brain with respect to different functional and anatomical regions of the microvascular tree. This may be of particular importance in clinical research as S1PR1 modulation is currently used in FDA-approved therapies in humans (*Brinkmann et al., 2010*).

From among the exogenous tracers typically used in BBB paracellular permeability studies, we chose three molecules with different sizes, NaFluo and Alexa Fluor 488 that readily diffuse via the BBB, and FITC-dextran that has low BBB permeability (*Shi et al., 2014*; *Kutuzov et al., 2018*).

The loss of BBB integrity in *Apom*[-/-] mice was observed for small tracer molecules < 0.7 kDa, but no significant leak was observed for 10 kDa dextran, which is in accordance with previous findings in S1PR1–knock-out mice (*Yanagida et al., 2017*). However, in contrast to Yanagida and colleagues (*Yanagida et al., 2017*), we show that it is not the complete absence of brain endothelial S1PR1 signaling, but a lack of a specific fraction of S1P circulating in the blood (i.e., apoM-bound S1P) is sufficient to impair the BBB, and occurs despite the presence of active S1PR1 (*Christoffersen et al., 2011*). Although we cannot exclude that some NaFluo molecules were entrapped in transport vesicles during their formation at the luminal side of the endothelium, in our experiments we did not detect any vesicular fraction of NaFluo, Alexa Fluor 488 and FITC-dx at the BBB. In contrast, albumin-conjugated Alexa 488 showed clear vesicular pattern, which is consistent with transcytosis (*De Bock et al., 2016*; *Minshall et al., 2000*; *Minshall et al., 2002*). Furthermore, NaFluo is a strong target for endothelium efflux pumps, that is multidrug resistance proteins (MRP) that sequester the tracer back to the blood stream (*Hawkins et al., 2007*). Therefore, we consider that the incidental transcytosis of NaFluo, Alexa Fluor 488 or FITC-dx had negligible contribution to fluorescence increases, and the effect observed in the brain parenchyma occurred due to the increase in paracellular permeability.

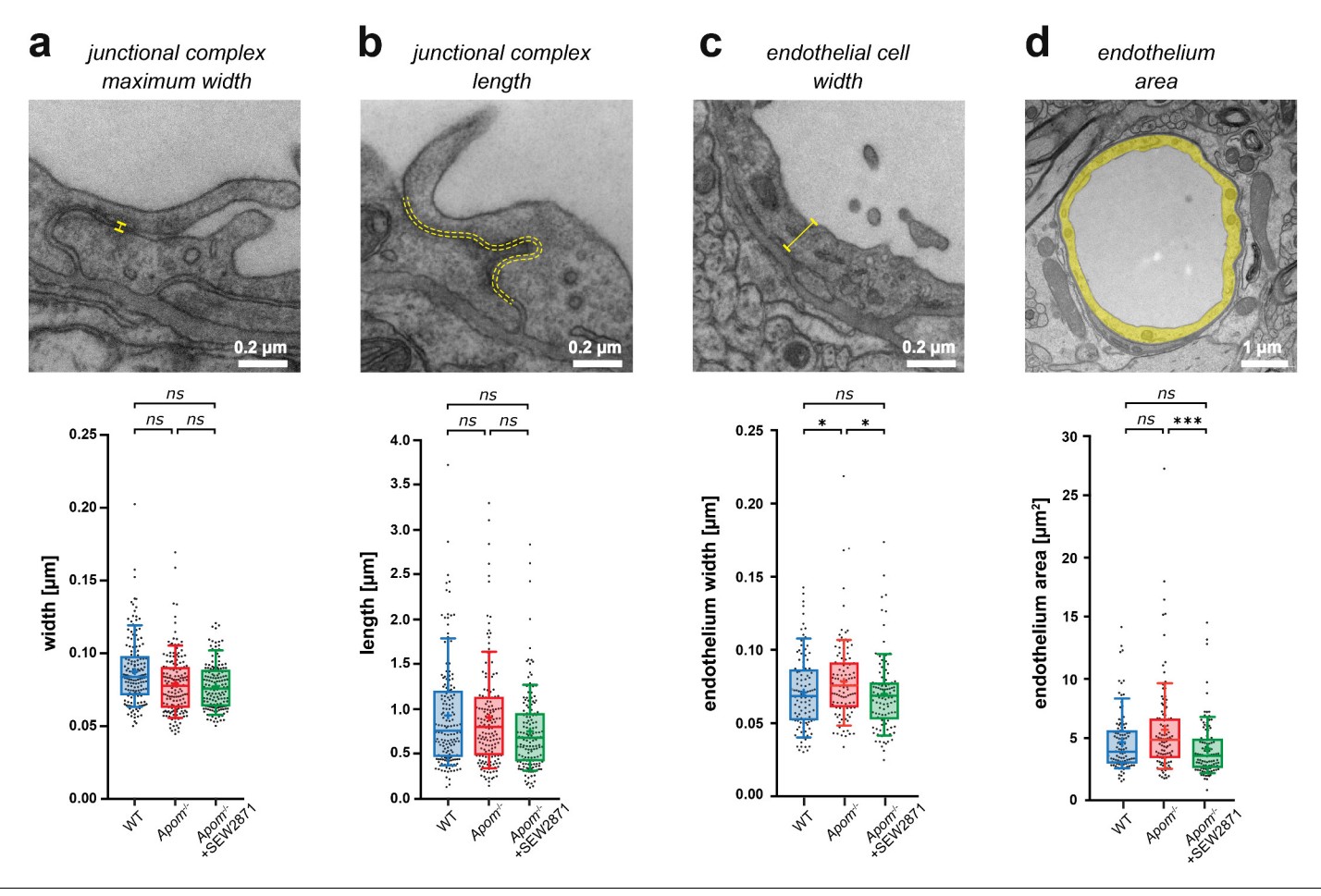

**Figure 2.** Small changes in endothelium cell morphology, but not junctional complexes, accompany increased BBB paracellular permeability in *Apom*⁻/⁻ mice. TEM ultrastructure assessment of the structural components of the BBB revealed no changes in the (a) width or (b) length of junctional complexes, with (c) a small increase in endothelial thickness in *Apom*⁻/⁻ mice compared to WT mice and a decrease in SEW2871-treated *Apom*⁻/⁻ mice. (d) A small decrease in endothelial cell area was observed upon SEW2871 treatment in *Apom*⁻/⁻ mice. Data are presented as interquartile distributions with median (horizontal line) and mean (plus sign). The box whiskers indicate 10–90% percentiles. *$p<0.05$, ***$p<0.001$.

The online version of this article includes the following figure supplement(s) for figure 2:

**Figure supplement 1.** Complementary BBB measurements by TEM.

The increased permeability along the paracellular pathway occurs most commonly in response to severe damage (*Farrell and Shivers, 1984*; *Houthoff et al., 1982*; *Lossinsky et al., 1995*) and is typically reported to be secondary to the ultrastructural abnormalities of junctional complexes, such as truncated morphology during acute BBB breakdown, for example in stroke, sepsis, and brain tumors (*Davies, 2002*; *Engelhardt et al., 2014*; *Knowland et al., 2014*); and in diabetes (*Salameh et al., 2016*). S1P has been previously shown as an endothelial barrier-enhancing molecule in the lungs (*Schaphorst et al., 2003*; *Dudek and Garcia, 2001*; *Garcia et al., 2001*) and recently, in the brain (*Yanagida et al., 2017*; *Swendeman et al., 2017*). Our data ties apoM shortage-induced increase in BBB permeability with S1PR1. The mechanisms of S1PR1-mediated protective effect are not yet fully understood, but may be linked with Rho superfamily of small GTPases. Stimulation of S1PR1 and S1PR2 activates preferentially Rac and Rho, respectively, whilst S1PR3 stimulation activates both GTPases (*Xiong and Hla, 2014*; *Windh et al., 1999*). Rac and Rho modulate cytoskeleton dynamics in various cell types (*Burridge and Wennerberg, 2004*), including the formation and maintenance of a cortical actin ring that stabilizes intercellular junctions between adjoining endothelial cells (*Garcia et al., 2001*; *McVerry and Garcia, 2004*). At physiological concentrations, S1P activates preferentially Rac via S1PR1 over Rho (*Garcia et al., 2001*; *Shikata et al., 2003*; *Hla, 2003*),

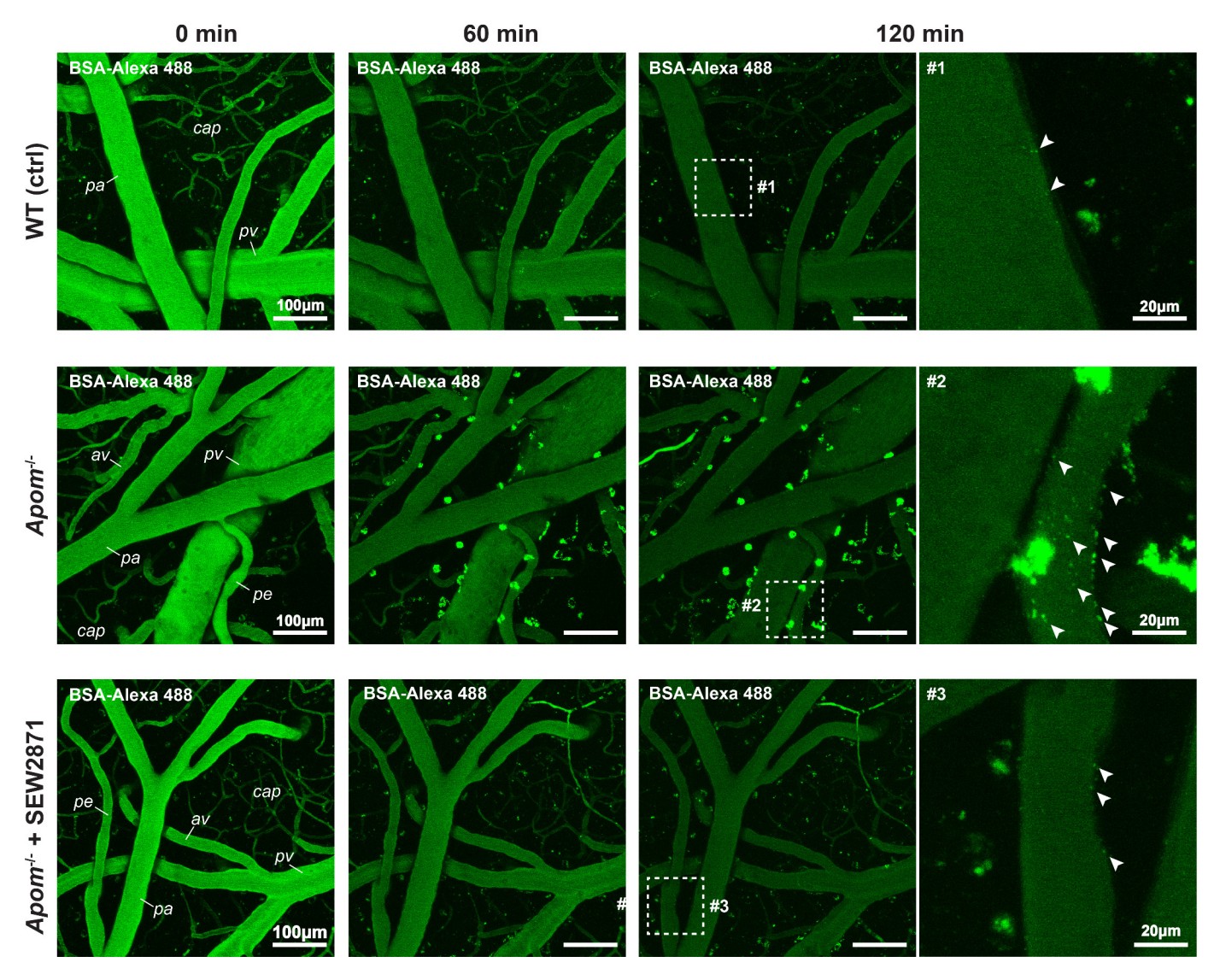

**Figure 3.** Increase in BSA-Alexa 488 uptake indicates an increase in transcellular transport in *Apom*[-/-] mice. Two-photon assessment of albumin uptake (green puncta) 60 and 120 min post i.v. injection of BSA-Alexa 488. Arrowheads indicate numerous puncta of BSA-Alexa 488 vesicles at the BBB interface in *Apom*[-/-] mice but only sparse labeling in WT and SEW2871-treated animals. pa = pial arteriole; pe = penetrating arteriole; cap = capillaries; pv = pial venule; av = ascending venule.

The online version of this article includes the following figure supplement(s) for figure 3:

**Figure supplement 1.** Increased BBB permeability in *Apom*[-/-] mice is not associated with fibrinogen invasion and gliosis.

**Figure supplement 2.** Uptake of BSA-Alexa488 in the *Apom*[-/-] brain parenchyma by macrophages.

which causes a redistribution of proteins that facilitate cortical actin ring assembly, and consequently tightening of the barrier (*Garcia et al., 2001*; *Mehta et al., 2005*). Inhibition of Rac downstream of S1PR1 leads to actin depolymerization, formation of F-actin stress fibers and increased permeability (*Wójciak-Stothard et al., 2001*; *Singleton et al., 2005*), and S1PR2 causes Rho activation that inhibits Rac (*Dudek and Garcia, 2001*; *Hla, 2003*). However, the formation and disassembly of actin cortical ring affecting BBB permeability should not be simply viewed as Rac vs. Rho antagonism, as inhibition of Rho, similarly to Rac, can reportedly also increase barrier permeability, although without interrupting the actin ring assembly (*Garcia et al., 2001*; *Etienne-Manneville and Hall, 2002*). Our quantitative TEM assessment shows that apoM deficiency did not induce changes in junctional complexes ultrastructure. This suggests that the increase of the BBB permeability in *Apom*[-/-] mice was

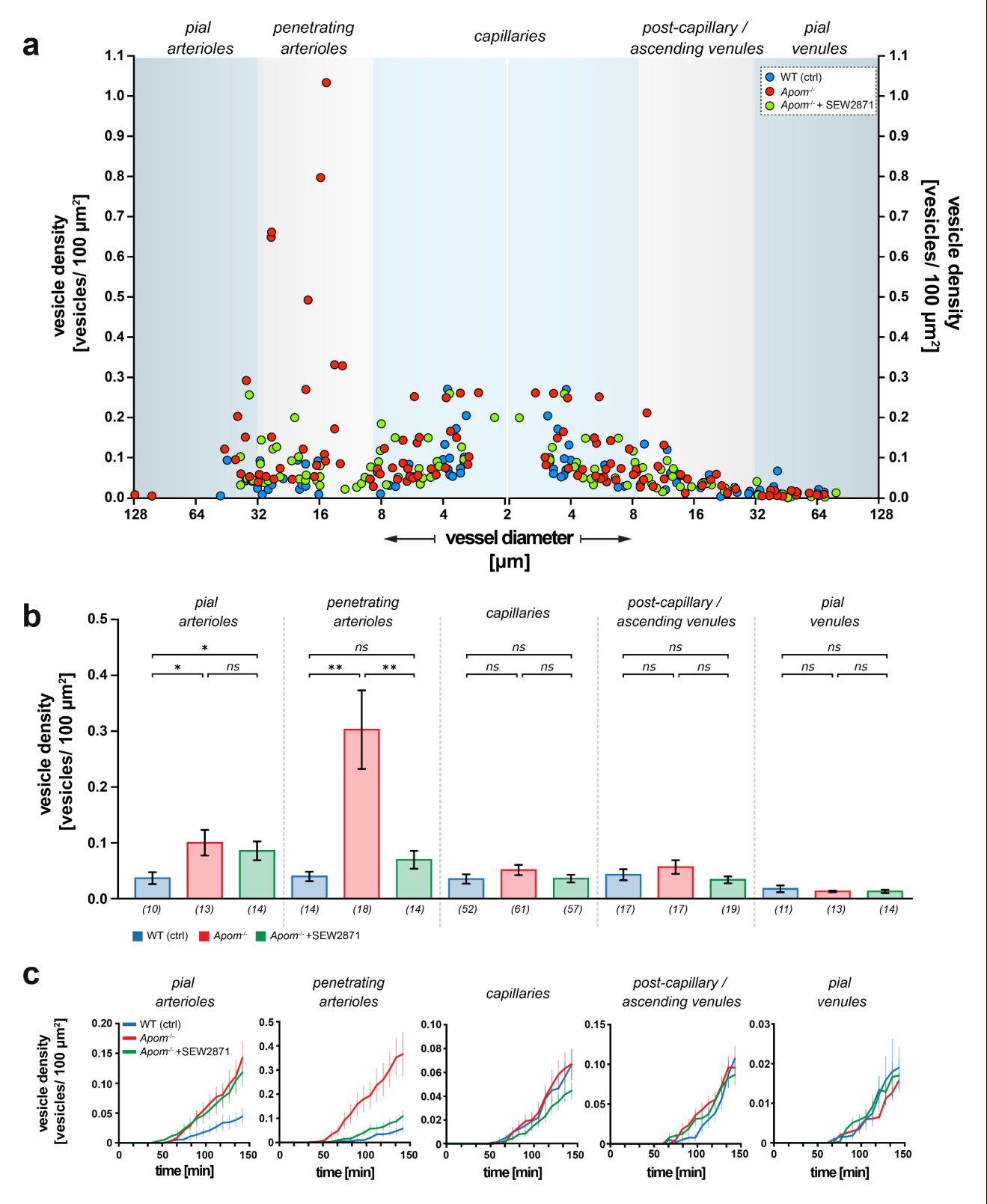

**Figure 4.** Heterogeneous susceptibility of different categories of brain microvessels to increased transcytosis in *Apom*⁻/⁻ mice. (**a**) Scatter plot showing the distribution of the surface vesicle densities of individual vessels plotted against respective vessel diameters 120 min post BSA-Alexa 488 injection. The blue areas represent the morphological division into distinct vessel types. Data from capillaries were mirrored to preserve the order of vessel diameter change in the continuous brain microvascular tree. (**b**) Quantitative analysis showed the increase in albumin uptake in *Apom*⁻/⁻ mice that was

*Figure 4 continued on next page*

*Figure 4 continued*

present only in the arterial part of the vascular tree and absent in capillaries and venules. Only penetrating arterioles responded to SEW2871 treatment. Numbers in brackets are the number of vessels across five mice in each group. The data are presented as mean ± SEM. (c) Average traces showing the kinetics of the vesicle density changes for WT, *Apom*<sup>-/-</sup>, and *Apom*<sup>-/-</sup> SEW2871 treated mice with respect to different vessel types. The data are presented as mean ± SEM. *p<0.05, **p<0.01.

either uncoupled from cortical actin ring remodeling, or that during S1PR1 hypostimulation and ongoing S1PR2-mediated Rho inhibition of Rac, the activity of Rac was still sufficient to maintain the elements of scaffolding complex of cortical actin ring that stabilize junctional complexes.

The small (~10%) increase in endothelium thickness in *Apom*<sup>-/-</sup> mice may be due to the impaired S1PR1 -dependent formation and the maintenance of the cortical actin ring; and formation of actin stress fibers facilitated by S1PR2 activation (*Garcia et al., 2001*; *Mehta et al., 2005*; *Wójciak-Stothard et al., 2001*; *Singleton et al., 2005*). Indirectly, BBB permeability changes may be due to osmotic imbalance and cell swelling, but it is uncertain whether this occurs in the brains of *Apom*<sup>-/-</sup> mice. Lack of S1PR1 activation aggravates edema in acute brain and lung injury (*Ito, 2019*), but here the system may not be challenged enough to induce osmotic swelling, as *Apom*<sup>-/-</sup> mice have no signs of lung edema and exhibit normal microvasculature morphology (*Christoffersen et al., 2011*). Other processes may also influence endothelium thickness, but their relevance in *Apom*<sup>-/-</sup> mice is yet to be established. For example, S1PR1 activity counteracts vascular endothelial growth factor (VEGF) -mediated endothelium contraction in angiogenesis (*Gaengel et al., 2012*), but whether S1PR1 hypostimulation affects VEGF signaling in the adult brain in *Apom*<sup>-/-</sup> mice is unknown. In addition, impaired S1PR1 signaling activates neuroinflammatory factors (*Blaho and Hla, 2014*; *Blaho et al., 2015*), and for example TNF-alpha has been shown to alter cytoskeleton and hyperplastic properties of endothelial cells (*Kang et al., 2008*; *Stroka and Aranda-Espinoza, 2011*). Thus, it cannot be excluded, that the changes in endothelium morphology result from both: direct effect of S1PR1 hypostimulation on endothelial cell, and indirect effects linked with impaired BBB-induced loss of brain homeostasis.

Notably, a significant increase in paracellular permeability without apparent changes in junctional complex morphology has been observed by others following osmotic shock (*Farrell and Shivers, 1984*) and in claudin-5-deficient mice (*Nitta et al., 2003*). The loss of BBB integrity in *Apom*<sup>-/-</sup> mice appeared to be to a lesser extent than in acute pathology (e.g., traumatic brain injury or ischemic stroke), as the BBB permeability towards 10 kDa dextran did not increase in *Apom*<sup>-/-</sup> mice (*Habgood et al., 2007*; *Özen et al., 2018*). Yet, even with is morphology intact, the BBB with a persistent increase in permeability may still contribute to brain pathology (*Carvey et al., 2009*).

Our in vivo results characterize apoM as a blood-borne molecule that can affect the BBB permeability via S1PR1. In inflammatory conditions, elevated concentrations of blood-circulating thrombin activate protease-activated receptor-1 (PAR1), and subsequently barrier-disruptive metalloproteinases (*Bogatcheva et al., 2002*; *Rempe et al., 2016*). On the contrary, low thrombin concentrations activate protein C (APC), a substrate for endothelial cell protein C receptor (EPCR), which alters PAR1 signaling to promote the synthesis of S1P via Sphk-1, and facilitates barrier enhancement (*Riewald et al., 2002*; *Feistritzer and Riewald, 2005*). In addition, APC has been shown to also induce S1PR1 phosphorylation, which further promotes the S1PR1 signaling branch in the endothelium (*Lee et al., 2001*). In line with our results in *Apom*<sup>-/-</sup> mice, the barrier enhancement by blood-borne thrombin occurred via S1PR1, as silencing of S1PR1 but not e.g. S1PR3 abolished the APC-mediated effect (*Feistritzer and Riewald, 2005*; *Finigan et al., 2005*). Except for apoM and thrombin, S1P signaling can be modulated by hyaluronic acid (HA). Depending on the form, i.e. high-molecular weight-HA (HMW-HA) or low-molecular weight (LMW-HA), the complex promotes interaction of a cell-surface glycoprotein CD44 isoforms with either S1PR1 or S1PR1/3, respectively (*Singleton et al., 2006*). This leads to enhancement of the barrier function via S1PR1 or an increase in BBB permeability via S1PR3 due to transactivation (i.e. phosphorylation) of respective receptors by CD44 isoforms, independently of S1P levels (*Singleton et al., 2006*). This further corroborates our findings on the regulation of S1P signaling by blood circulating molecules via preferential activation of specific S1P receptors.

However, the analogies should be made with caution, as the mechanistic insights come primarily from in vitro barrier models or non-brain tissue, where endothelial barrier properties may differ from

the BBB in vivo. For example, Rac and Rho are known to elicit cell-type and subcellular localization-specific effects (*Burridge and Wennerberg, 2004*; *McVerry and Garcia, 2004*; *Donati and Bruni, 2006*), and their intracellular pathways are cross-linked with numerous regulatory feedback loops, including for example Rho inhibiting Rac and vice versa (*Burridge and Wennerberg, 2004*; *Sander et al., 1999*). This adds to the complexity of interpretation of *Apom* knock-out effects on the BBB and is currently an active area in research.

The hypostimulation of S1PR1 in *Apom*$^{-/-}$ mice is likely to shift the signaling equilibrium towards cellular pathways that promote activation of Rho (via S1PR2) over Rac (via S1PR1), and the effects of increased BBB permeability are likely to be the net outcome of the failure in homeostatic interplay between receptor-dependent branches of S1P signaling, rather than simply a decrease in Rac activity. Consequently, the observed impairment of the BBB cannot simply be linked to a reduction in the amount of circulating S1P, but more specifically to the lack of apoM-bound S1P in favor of albumin-bound S1P in the blood. This notion is further supported by the result of the selective stimulation of S1PR1, which restored the BBB permeability in *Apom*$^{-/-}$ mice to the level of healthy controls.

Given that inhibition of S1PR2 tightens the endothelial barrier in lungs by facilitating cortical actin ring assembly and stabilization of tight junctions (*Sanchez et al., 2007*), and in the brain supports the BBB functional integrity (*Kim et al., 2015*; *Cao et al., 2019*), it is plausible that S1PR2 stimulation may alleviate BBB phenotype in *Apom*$^{-/-}$ mice. It should be noted, however, that S1PR1 expression in the endothelium is higher in the arterial branch of cerebral vasculature compared to venules, whereas S1PR2 expression is higher in venules than in arterioles (*Vanlandewijck et al., 2018*). It may lead, similar as in SEW2871 treatment, to region-specific effects, where the BBB responses to S1PR2 modulation may differ between distinct types of cerebral microvessels.

Though paracellular permeability restriction is typically viewed as the main functional constituent of the BBB, the low prevalence of transcytosis is equally important for preservation of the brain microenvironment (*Knowland et al., 2014*; *Habgood et al., 2007*; *Hashizume and Black, 2002*; *Krueger et al., 2013*). Recently, Mfsd2a was identified as a critical protein in the suppression of transcytosis. Similar to *Apom*$^{-/-}$, a lack of Mfsd2a leads to an increase in vesicular transport across the endothelium that compromises the BBB without structurally opening the paracellular route (*Ben-Zvi et al., 2014*) or changing junctional protein composition (*Yang et al., 2017*). Furthermore, the increase in BBB permeability can occur via both the transcellular and paracellular routes, but at different time points; e.g. the increase in transcytosis rate can precede paracellular leakage in stroke (*Knowland et al., 2014*), and selective dysregulation of the transcellular pathway can occur while the tight junctions are preserved (*Krueger et al., 2013*). Thus, the intact morphology of junctional complexes, as shown here, is not necessarily indicative of preserved BBB properties.

Noteworthy, the lack of apoM-S1P lead to an increased rate of transcytosis, though only in pial and penetrating arterioles. The susceptibility of the arterial branch of the microvasculature to apoM shortage may be linked to preferential expression of S1PR1 and APC in pial and penetrating arterioles (*Vanlandewijck et al., 2018*), thus possibly high reliance in these vessel types on S1PR1 signaling in the maintenance of the BBB. It is unclear, however, why penetrating, but not pial arterioles responded to SEW2871, but anatomical differences may be of importance. For example, S1P signaling promotes the formation of the glycocalyx, which limits the diffusion of hydrophilic and negatively charged substances, and supports the barrier function (*Zhang et al., 2016*). Glycocalyx in pial arterioles is estimated to be thicker compared to the penetrating arterioles (*Yoon et al., 2017*), but it is unknown how S1PR1 hypostimulation affects distribution and coverage of glycocalyx layer in *Apom*$^{-/-}$ mice. In addition, the reconstitution of barrier function may have a different time course in pial and penetrating arterioles. Lastly, in contrast to bolus SEW2871 injection, a sustained stimulation of S1PR1 may be needed to restore BBB function in pial arterioles during ongoing apoM deficit.

High-density lipoprotein (HDL) particles are heterogeneous in size and content of apolipoproteins and lipids. In humans, HDL plasma concentrations correlate with reduced susceptibility to cerebro- and cardiovascular diseases (*Rader and Tall, 2012*; *Hovingh et al., 2015*; *Fellows et al., 2015*), and approximately 5% of HDL contains apoM (*Xu and Dahlbäck, 1999*; *Christoffersen et al., 2006*). The beneficial effects of HDL can be, in part, explained by HDL association to apoM/S1P. We show that lack of apoM increases the flux of small molecules across the BBB, similar to previous findings investigating the permeability in lung and adipose tissue (*Christoffersen et al., 2011*; *Christensen et al., 2016*; *Christoffersen et al., 2018*). Stressing the system even further, lack of apoM increases the susceptibility to neuroinflammation (*Blaho et al., 2015*; *Galvani et al., 2015*).

Mainly this has been interpreted as effects caused by blood-circulating apoM/S1P complex. However, a recent study suggested that porcine brain endothelial cells express and secrete apoM into the brain parenchyma (*Kober et al., 2017*). This has not been reported in other species but could also play a role in the increased permeability observed in the present study. Except S1P, apoM can potentially bind other ligands, for example retinoic acid (*Ahnström et al., 2007*) or oxidized phospholipids (*Elsøe et al., 2012*). Lack of apoM would increase free oxidized phospholipid content, which potentially can also affect the BBB (*Singleton et al., 2009*). However, reversing the BBB phenotype by SEW2871 suggests that the observed effects are primarily mediated via S1PR1 rather than for example oxidized phospholipids. Noteworthy, with the absence of apoM, S1P may still associate to HDL. A recent study showed that in mice lacking expression of both, apoM and albumin, S1P binds to HDL via novel S1P binding protein apoA4 (*Obinata et al., 2019*). This may potentially constitute a compensatory mechanism in $Apom^{-/-}$ mice, but whether HDL/apoA4/S1P complex also plays a role in maintaining the BBB is unknown and will need further investigations.

Two-photon imaging allows studying the endothelial barrier in the living brain with all its structural and functional constituents, including blood flow, and at the resolution of single microvessels. However, like any imaging approach, it has also limitations that apply to our study. In contrast to transcellular vesicular transport, the fast diffusion of NaFluo in the brain parenchyma made it not feasible to analyze for differences in paracellular permeability increase between distinct vessel types in $Apom^{-/-}$ mice. Regarding the transcellular permeability assessment, we have only investigated the transcytosis of albumin, which represents a fraction of receptor-mediated transcellular pathways. Lastly, the surgical procedures might also affect the brain physiology, however in our previous works, we observed no significant effects of cranial window preparation and imaging on the neurovascular coupling, neuronal and astrocytic calcium transients, electrophysiological activity and cell morphology, suggesting lack of effects on the brain that could significantly affect the BBB function (*Kutuzov et al., 2018*; *Kucharz and Lauritzen, 2018*; *Hall et al., 2014*). Instead of acute craniotomy, further studies may employ chronic window imaging to address changes in BBB permeability in $Apom^{-/-}$ mice over a longer time.

In summary, apoM is critical for maintaining low paracellular and transcellular permeability at the BBB. The BBB is heterogeneous in the susceptibility towards apoM shortage and in the potential for rescue mediated by the S1PR1 signaling pathway. The pronounced increase in transcytosis in $Apom^{-/-}$ mice stresses the importance of considering the development of protective strategies for this pathway to regulate the immediate microenvironment of brain cells and reduce secondary neural damage in both acute and chronic neurological conditions.

## Materials and methods

### Animals

All animal procedures were approved by the Danish National Ethics Committee and were in appliance with the ARRIVE guidelines. We used 14 to 20 week-old (22–28 g) female transgenic C57bl/6 apoM-deficient mice ($Apom^{-/-}$) (*Christoffersen et al., 2011*), and age- and gender-matched wild-type (WT) mice as the control group.

### Anesthesia

Anesthesia was induced by a single bolus intraperitoneal (i.p.) xylazine (10 mg/kg body weight [BW]) and ketamine (60 mg/kg BW) injection. During all steps of the surgical preparation, the anesthesia was maintained by i.p. injections of ketamine (30 mg/kg BW) at 25 min intervals. Thirty minutes before imaging, the anesthesia was changed to continuous intravenous (i.v.) infusion of alpha-chloralose (50 µg/hour/g BW).

### Animal preparation for two-photon imaging

Animals were operated on as described previously (*Kucharz and Lauritzen, 2018*). Briefly, a tracheotomy was performed for mechanical respiration (MiniVent Type 845, Harvard Apparatus) and to monitor exhaled $CO_2$ (MicroCapstar End-tidal $CO_2$ Analyzer, CWE). Two catheters were inserted: the first into the left femoral artery to monitor arterial blood pressure (Pressure Monitor BP-1, World

Precision Instruments) and inject compounds, and the second into the femoral vein for infusion of the anesthetic.

Next, a craniotomy was performed. The exposed skull was attached to a custom-made metal head bar with cyanoacrylate gel (Loctite Adhesives) and mounted onto the microscope imaging stage insert. The opening was made using a diamond dental drill (at 7000 rpm) over the somatosensory cortex (4 mm diameter; coordinates: 3 mm lateral, 0.5 mm posterior to bregma). During the procedure, the craniotomy and drill were cooled to room temperature repeatedly with artificial cerebrospinal fluid (aCSF; in mM: NaCl 120, KCl 2.8, $Na_2HPO_4$ 1, $MgCl_2$ 0.876, $NaHCO_3$ 22, $CaCl_2$ 1.45, glucose 2.55, at 37°C, pH = 7.4) to prevent brain damage from excessive heating of the drill. The dura was removed and a drop of 0.75% agarose (type III-A, Sigma-Aldrich) applied onto the craniotomy. The opening was then secured with an imaging coverslip (Menzel-Gläser) and kept under aCSF to preserve humidity. To ensure physiological $O_2$ and $CO_2$ blood concentrations, prior to imaging, a 50 µl blood sample was collected via the arterial catheter (ABL blood gas analyzer, Radiometer), and the respiration rate and volume adjusted if necessary.

## Fluorescent probes

FITC-dextran (MW 10 kDa, 0.5%, 0.05 ml, Sigma-Aldrich), Alexa Fluor 488 (MW 0.643 kDa, 0.025%, 0.04 ml, Invitrogen), and sodium fluorescein (NaFluo, MW 0.365 kDa, 1%, 0.04 ml, Sigma-Aldrich) were each dissolved in sterile saline and administered via the arterial femoral catheter as single bolus injections. Bovine serum albumin (BSA) Alexa Fluor 488-conjugate (BSA-Alexa 488, Invitrogen) was administered via the arterial femoral catheter as a single bolus injection (0.05 ml 1%), together with TRITC-dextran (MW 65 kDa, 0.05 ml 1%, Sigma-Aldrich) for delineation of the vasculature and to ascertain that vessels were not damaged following the cranial window microsurgery that could lead to false-positive results.

## Two-photon imaging setup

Two-photon imaging was performed using an SP5 upright laser scanning microscope (Leica Microsystems) equipped with a MaiTai Ti:Sapphire laser (Millennia Pro, Spectra-Physics) using the 20 × 1.0 NA water-immersion objective. The emitted light was split by a FITC/TRITC filter and collected by separate multi-alkali detectors after 525–560 nm and 560–625 nm bandpass filters (Leica Microsystems).

## Monitoring paracellular permeability

The excitation wavelength for FITC-dextran, Alexa Fluor 488, and NaFluo was 850 nm (14 mWatt output power at the sample). Data were collected as 16-bit image hyperstacks at 512 × 512 pixel resolution (775 µm x 775 µm) in bidirectional scanning mode with a Z-step size of 5 µm (depth span = 114 µm) and 1 min time interval between Z-stacks. The total recording time for each fluorophore was 30 min. The recordings were exported to ImageJ (NIH, version 1.48u4) and Z-projected using maximal signal intensity projection. Regions of interest (ROIs) were chosen throughout the brain parenchyma at all locations deprived of vasculature for the measurement of fluorescent intensity. An increase in fluorescent intensity over the baseline (first projected Z-stack in the time-lapse recording) in the parenchyma indicated crossing of the blood-circulating fluorophore into the brain.

FITC-dextran, Alexa Fluor 488, and NaFluo were consecutively injected in 30 min intervals into the same animal. We did not wait for the fluorophores clearance, as the time of bloodstream clearance differs between fluorophores, and varies between animals. Consequently, this could obscure characterization of the biological effects of SEW2871 on the BBB, as the assessment of permeability in each mouse for each respective tracer would be performed at different time after the drug administration. SEW2871 has a relatively short half-time in the blood plasma with its biological effects peaking at 1 hr, and returning to baseline 4.5 hr after administration (as determined by, for example Akt phosphorylation)(*Jo et al., 2005*). Instead, we chose to standardize the time of imaging after SEW2871 injection, that is FITC-dx, Alexa Fluor 488, and NaFluo were always imaged 2.5 hr, 3 hr, 3.5 hr after SEW2871 administration, respectively. In our calculations we estimated the contribution of a preceding fluorescence accumulation to a subsequent fluorophore(s) signal measurement. For each mouse, we subtracted the constant autofluorescence background from FITC-dx, Alexa Fluor 488, and NaFluo trace and performed an exponential fit to the FITC-dx trace, and the fit was

extrapolated to subsequently imaged Alexa Fluor 488 and NaFluo (*Figure 1—figure supplement 1a*). We chose the exponential fit to account for the largest systematic error in our experiment, that is the largest theoretical contribution of a preceding tracer to a subsequent tracer signal. The fluorescence is additive, therefore we corrected Alexa Fluor 488 and NaFluo traces by subtracting FITC-dx extrapolated fit values (*Figure 1—figure supplement 1a*) and effectively, uncoupled FITC-dx kinetics from Alexa Fluor 488 and NaFluo. Next, to the corrected trace of Alexa Fluor 488 we performed an exponential fit and extrapolated it to the time of NaFluo imaging. Similarly as before, we subtracted Alexa Fluor 488 extrapolated fit from NaFluo trace (*Figure 1—figure supplement 1b*). The same approach, but with exponential decay fit, was used to account for the baseline fluorescence changes in the bloodstream (*Figure 1—figure supplement 1c–d*).

## Monitoring transcellular transport

The animals were injected intravenously with BSA-Alexa 488. The excitation wavelength was 870 nm (17 mWatt output power at the sample). Data were collected as 16-bit hyperstacks at 2048 × 2048 pixel resolution (386.66 μm x 386.66 μm) in bidirectional scanning mode and triple frame averaging with a Z-step size of 2.50 μm, depth span 144 μm, and 7.5 min interval between Z-stacks for a total recording time of 120 min. The data were exported to ImageJ and Z-projected using maximal signal intensity projection. Based on morphology and location, the vessels were classified as pial arterioles or venules, penetrating arterioles, post-capillary venules, ascending venules, or capillaries. To determine the vesicle surface density, the vessel surface area was calculated based on the diameter and length of the vessel from which the BSA-Alexa 488 signal was counted.

## Transmission electron microscopy

The mice were anesthetized with bolus i.p. injections of xylazine (10 mg/kg BW) and ketamine (60 mg/kg BW), then perfused with Karnovsky fixation (2% glutaraldehyde and 4% paraformaldehyde). Brains were harvested and the cortex cut into 1 mm cubes and left for further fixation in 2% glutaraldehyde in 0.05 M phosphate buffer for 7 days. For transmission electron microscopy (TEM) staining, the samples were washed three times for 15–20 min in 0.12 M cacodylate buffer, followed by post-fixation in 1% osmium tetroxide ($OsO_4$) and 0.05 M (1.5%) potassium ferricyanide ($K_3FeCn_6$) in 0.12 M cacodylate buffer for 1 hr at room temperature (RT). Next, the samples were washed three times for 15–20 min in ddH$_2$O and then dehydrated in an ethanol gradient (70%, 96%, and 3 × 15 min in absolute ethanol). Infiltration was performed with propylene oxide twice for 15 min, followed by incubation for 20–40 min in a resin epon/propylene oxide (Bie and Berntsen) gradient (1:3, 1:1, 3:1) and pure resin for 2 hr. Samples were embedded in resin at 60°C for 24 hr. Sections were cut at a thickness of 70 nm (Leica EM UC7), placed on copper grids (Leica), and stained with lead citrate (Leica Ultrostain II, Leica Microsystems) for 3 min and uranyl acetate (Leica Ultrostain I, Leica Microsystems) for 10 min using an automatic contrasting system (Leica EM AC20, Leica Microsystems). Imaging was performed using a CM100 transmission electron microscope (Phillips/FEI, ITEM software) at 9700x – 17500x and 46000-60000x magnification. Two samples from each mouse were imaged, and 20–30 capillaries from each sample were chosen randomly. ImageJ was used for the measurement of BBB morphology.

## Immunohistochemistry

The isoflurane-anesthetized mice were transcardially perfused at the rate of 10 ml/min for 1 min with saline and then for 7 min with ice-cold 4% paraformaldehyde (PFA). The brains were placed in 4% PFA for 24 hr at 4°C for post-fixation, then in 25% sucrose in PBS for ~ 24 hr at 4°C. The brains were frozen in dry ice and cut into a series of 35 μm thick coronal sections (Microm HM450 microtome, Thermo Fisher Scientific).

For fibrinogen staining, we performed antigen retrieval in Tris/EDTA buffer (pH 9.0) for 30 min at 80°C, with a quenching step in 3% $H_2O_2$ and 10% methanol in Tris-buffered saline (TBS) for 20 min. The sections were washed 3x in TBS and pre-incubated for 1 hr at the RT with 5% normal goat serum in 0.25% triton-X in TBS (TBS-T). Subsequently, the sections were incubated o/n at RT with anti-fibrinogen primary antibody (1:2000, ab227063, Abcam) in 5% normal goat serum TBS-T solution. Following that, the tissue sections were rinsed 3 × 10 min in TBS-T and incubated in 1% BSA TBS-T with biotinylated secondary goat-anti-rabbit antibody (1:200; Vector Laboratories Inc). Next, the

sections were washed 3x in TBS-T for 10 min, then incubated with an avidin-biotin-peroxidase complex solution for 1 hr at RT, and washed again 3 × 10 min with TBS-T. The staining was visualized using 3,3'-diaminobenzidine (DAB) and 0.01% $H_2O_2$ according to manufacturer's instructions. Lastly, the sections were mounted on chromatin-gelatin coated glass slides, stained in 0.1% cresyl violet (CV) solution for 3 min, dehydrated in increasing alcohol solutions (70%, 95%, 100%), cleared in xylene and using a DPX mountant (Sigma-Aldrich), the slides were coverslipped for imaging.

Heat-mediated antigen retrieval was performed on the free-floating brain sections using sodium citrate buffer (pH6) for anti-TMEM119 staining, and EDTA buffer (pH 8.0) at 80°C for 30 min for anti-macrophage staining (RM0029-11H3). The sections were rinsed 3 × 10 min in 0.1 M potassium phosphate-buffered saline (KPBS), then placed for 1 hr in KPBS 0.05% Triton-X (KPBS-T) containing 5% normal goat serum. Next, the sections were incubated o/n at RT with primary antibodies in 5% normal goat serum solution in KPBS-T. The primary antibodies used were: anti-macrophage (1:200; RM0029-11H3; rabbit; Abcam cat# ab56297), recombinant anti-TMEM119 (1:200; rabbit; Abcam cat# ab209064), anti-Iba1 (1:1000; rabbit; Wako cat# 019–19741) and anti-GFAP (1:500; rabbit; DAKO cat# Z033429-2). The sections were rinsed 3 × 10 min in KPBS-T and then incubated with Alexa647-conjugated secondary antibody (1:200, goat-anti-rabbit, Invitrogen) in KPBS-T for 2 hr at RT. Following that, the sections were rinsed 3x for 10 min with KPBS, and incubated with DAPI (4',6-diamidino-2-phenylindole) solution (Thermo Fisher Scientific) for 10 min. Lastly, the sections were mounted on gelatin-coated slides and secured with coverslips using an anti-fading medium (SlowFade Diamond Antifade Mountant, S36963, Thermo Fisher Scientific) for fluorescence confocal imaging.

## Confocal imaging

Confocal imaging was performed using DMi8 inverted fluorescence microscope (Leica Microsystems) equipped with a 40 × 1.3 NA oil-immersion objective. DAPI, BSA-Alexa488, and Alexa647 were excited with 405 nm, 488 nm, and 638 nm laser diodes, respectively. The imaging was performed in sequential mode (each channel separately), to avoid fluorescence bleed-through between the channels. The emitted light was collected by a hybrid detector after: 410–450 nm (for DAPI), 510–550 nm (for BSA-Alexa488), and 650–750 nm (for Alexa647–conjugated antibodies) bandpass.

Bright-field imaging of fibrinogen/cresyl violet staining was performed using BX53 upright transmitted light microscope (Olympus) equipped with a 60 × 1.4 NA oil-immersion objective.

All images were exported to ImageJ (NIH, version 1.48u4) for further analysis. For comparisons between WT and $Apom^{-/-}$ mice, the images were collected using the same microscope settings, and Z-projected using maximal signal intensity projection from an equally thick section (=10 μm). The WT vs. $Apom^{-/-}$ mice imaging data on the figures are presented in the fluorescence signal intensity scale.

## Drug administration

SEW2871 (Sigma-Aldrich) was diluted in PBS (to 0.1 mg/ml in 1% DMSO) and administered as a single i.p. bolus injection (10 μg/g BW) 150 min prior to two-photon imaging or perfusion fixation.

## Statistical analysis

Statistical analysis was performed using Prism 7 (v7.0b, Graphpad) and R software (v3.5.1). The data were tested for normal distribution with Pearson's normality test and either an unpaired two-tailed Student's t-test with Welch's correction (normally distributed data) or Mann-Whitney test (non-normally distributed data) was used. The TEM data underwent logarithmic transformation and was evaluated using one-way analysis of variance (ANOVA) followed by Tukey's multiple comparison test. The sample size has been selected based on our previous experiments using two-photon in vivo microscopy (*Kucharz and Lauritzen, 2018*), in vivo electrophysiology (*Kucharz et al., 2017*) and TEM (*Christoffersen et al., 2011*). The data consisted only of biological replicates, the number of independent biological replicates (N) is provided in the results section and/or on figures. All analyses were performed in randomized order, with TEM analysis performed blinded to experimental conditions. No animals or data outliers have been excluded from analysis.

## Acknowledgements

We thank the Core Facility for Integrated Microscopy at the University of Copenhagen, especially Klaus Qvortrup, for sample preparation for TEM. We thank Micael Lønstrup and Nikolay Kutuzov from the Department of Neuroscience at the University of Copenhagen for the assistance in the animal surgeries and comments on the manuscript, respectively. This work was supported by the Lundbeck Foundation, the Danish Council for Independent Research (Medical Sciences), and a Nordea Foundation Grant to the Center for Healthy Aging.

## Additional information

### Funding

| Funder | Grant reference number | Author |
|---|---|---|
| Lundbeck Foundation | RIBBDD Initiative R302-2018-2266 | Martin Lauritzen |
| Det Frie Forskningsråd | 0602-01965B | Martin Lauritzen |
| Nordea Foundation | Grant to the Center for Healthy Aging 114995 | Martin Lauritzen |

The funders had no role in study design, data collection and interpretation, or the decision to submit the work for publication.

### Author contributions

Mette Mathiesen Janiurek, Data curation, Formal analysis, Validation, Investigation, Visualization, Writing—original draft; Rana Soylu-Kucharz, Resources, Validation, Investigation, Visualization, Methodology, Writing—review and editing; Christina Christoffersen, Resources, Investigation, Writing—review and editing; Krzysztof Kucharz, Conceptualization, Formal analysis, Supervision, Validation, Investigation, Visualization, Methodology, Writing—original draft, Writing—review and editing; Martin Lauritzen, Conceptualization, Supervision, Funding acquisition, Writing—original draft, Project administration, Writing—review and editing

### Author ORCIDs

Krzysztof Kucharz (iD) https://orcid.org/0000-0002-6852-5300

### Ethics

Animal experimentation: All animal procedures were approved by the Danish National Ethics Committee (permit no. 2019-15-0201-01655) and were in appliance with the ARRIVE guidelines.

### Decision letter and Author response

Decision letter https://doi.org/10.7554/eLife.49405.SA1
Author response https://doi.org/10.7554/eLife.49405.SA2

## Additional files

### Supplementary files

• Transparent reporting form

### Data availability

Dataset has been deposited at Dryad Digital Repository: https://doi.org/10.5061/dryad.qg04542.

The following dataset was generated:

| Author(s) | Year | Dataset title | Dataset URL | Database and Identifier |
|---|---|---|---|---|
| Krzysztof Kucharz. Mette Mathiesen | 2019 | Data from: Apolipoprotein M-bound sphingosine-1-phosphate | https://doi.org/10.5061/dryad.qg04542 | Dryad Digital Repository, 10.5061/ |

Janiurek, Rana Soylu-Kucharz, Christina Christoffersen, Martin Lauritzen

regulates blood-brain barrier paracellular permeability and transcytosis

dryad.qg04542

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
