## [Decision Letter]

**Acceptance summary:**

The authors examined the blood-brain barrier (BBB) function of WT and ApoM KO mice, and show using two-photon microscopy that ApoM KO mice exhibit a BBB defect for small molecules. Although this was expected to some degree from prior work, the present manuscript provides a beautiful demonstration using a state-of-the-art live animal imaging. The data are convincing and demonstrate that ApoM / S1P signaling via endothelial S1PR1 is important in controlling the BBB function of small molecular weight compounds.

**Decision letter after peer review:**

Thank you for submitting your article "Apolipoprotein M-bound sphingosine-1-phosphate regulates blood-brain barrier paracellular permeability and transcytosis" for consideration by *eLife*. Your article has been reviewed by Gary Westbrook as the Senior Editor, a Reviewing Editor, and three reviewers. The following individuals involved in review of your submission have agreed to reveal their identity: Timothy Hla (Reviewer #2). The reviewers have discussed the reviews with one another and the Reviewing Editor has drafted this decision to help you prepare a revised submission.

Summary:

As you will see from the original text of the reviews below, there were substantial differences in the reviewers' judgment on the potential impact of this work. We discussed this topic among the reviewers. There was consensus that the work was of high quality using a sensitive technique, but the following issues must be addressed in the revision along with a point-by-point response to ALL reviewers' comments. As you will see some of these points will likely require additional data or analysis.

1) There were places where better quantification is necessary (see point 1 of reviewer 1).

2) A more rigorous discussion of the limitations of the study and how it advanced what is already known is needed (see point 2 of reviewer 1).

3) A discussion of alternative hypotheses, e.g. other effects of germline apoM deletions or compensation by other HDL components should be included.

4) It is important to demonstrate whether there was a glial/microglia reaction following entry of plasma proteins (fibrin stain) into the brain parenchyma. (see point of reviewer 3).

5) Do the authors have data on EM analysis of arterioles and arteries (see point 3 of reviewer 1)?

*Reviewer #1:*

Sphingosine-1-phosphate (S1P) is a bioactive sphingolipid present in blood, where it is typically bound to either circulating apolipoprotein M (apoM; about 70%) or albumin (about 30%). S1P will preferentially activate different S1P G-protein coupled receptors depending on its binding partner in blood, and apoM-bound S1P preferentially binds to cell surface S1PR1 receptors. Here the authors studied the effect of apoM deficiency on blood brain barrier (BBB) permeability using in vivo two-photon microscopy of extravasated tracers and EM analysis of endothelial tight junctions. Knockout of apoM (apoM-/-) in mice induced increased extravasation of small dye molecules, which could be ameliorated by treatment with S1PR1 agonist SEW2871. EM imaging indicated no changes in endothelial junction length or width in capillaries, but in vivo two-photon microscopy revealed increased numbers of extra-vascular "vesicles" along the arterioles and arteries in the brains of apoM-/- mice, which could be reduced only in the penetrating arterioles with application of S1PR1 agonist SEW2871.

This manuscript presents interesting observations in an understudied subject. The manuscript would be improved by addressing the following:

Essential revisions:

1) The experiments in FigureFigure 1 show increased leakage of the small molecule NaFluo in apoM-/- mice (without quantification). The conclusion in the text indicating that this effect is due to paracellular permeability (subsection “ApoM / S1PR1 signaling deficit increases transcellular transport”) is premature. At this point, all that has been shown is a descriptive illustration of enhanced leakage, not "paracellular permeability." Also, from subsequent data presented in the manuscript, it would seem that transcellular transport would play a role in increasing NaFluo into the brain. This portion of the manuscript (subsection “ApoM / S1PR1 signaling deficit increases transcellular transport”) should be revised, quantified and/or possibly combined with the data in Figure 2.

2) These findings seem to be consistent with previous reports showing that activation of S1PR1 reduces endothelial permeability through rearrangement of cellular cytoskeleton and that S1P leads to barrier enhancement via S1PR1 (see for example McVerry and Garcia, 2004; Burridge and Wennerberger, 2004; Singleton.et al., 2005). Additionally, activated protein C enhances endothelial barriers via EPCR-dependent PAR1-mediated activation of Sphk-1 which generates sphingosine-1-phosphate S1P acting on S1PR1 (Feistritze and Riewald, 2005; Finigan, 2005). Please discuss how the authors observations fit with this previous literature.

3) The EM analysis of endothelial junctions is very nicely done, but performed only at the capillary level. Since the authors found that arterioles and arteries show the most pronounced changes in vesicular transport between genotypes with in vivo imaging, should these vessels be also checked for endothelial junction integrity in apoM-/- mice?

4) In the experiments presented in Figure 2, three tracers with similar fluorescence spectra are sequentially injected into the vasculature. Injection of one tracer after another of the same color has the potential to confound data of the following tracer injections. Have the authors checked that previous tracers injected have cleared before injecting the next tracer in these experiments? Is a baseline taken before each subsequent tracer injection?

5) In the BSA-Alexa 488 transcellular experiments illustrating vesicles shown in Figure 4, could the authors confirm the cell type taking up the BSA in the brain with histology?

6) Additionally, the BSA-Alexa 488 transcellular data would be strengthened by adding a conformation of the observations using a different method, e.g. immunostaining or EM imaging.

7) Why were the authors able to see the larger BSA-Alexa 488 tracer cross the BBB, but not the FITC-dextran?

8) The finding that the endothelial thickness is increased with apoM knockout is interesting. What might be the reason for this? Swelling or something else? Please discuss.

*Reviewer #2:*

Janiurek et al., studied the BBB function of WT and ApoM KO mice. They show, using two-photon microscopy, that ApoM KO mice exhibit a BBB defect in small molecules. This was expected since we know from studies of Yanagida et al., that S1PR1 ECKO also show this phenotype. The present manuscript provides a beautiful demonstration of this using state of the art, live animal imaging studies. The data are convincing and demonstrate that ApoM / S1P signaling via the endothelial S1PR1 is important in controlling the BBB function of small molecular weight compounds. The authors also examined the endothelial morphology and junctions as well as transcytosis in various vessel subtypes. They report different changes in transcytosis in WT and ApoM KO in penetrating arterioles vs capillaries.

Overall, the study is well done and impressive. Some of the results are expected since we already know that S1PR1 regulates the same function in the BBB. It would be useful if the authors were to extend the discussion they had with S1PR2. For example, does S1PR2 inhibitor inhibit the increased permeability seen in ApoM KO mice? Since they have the techniques to examine this issue, this would add much to the field.

There are some minor errors, such as G proteins downstream of S1PR1 and S1PR2 and not GPCRs. Also, a better discussion of HDL/ ApoM function and BBB would be of interest.

*Reviewer #3:*

The following observations are already published and generally acknowledged:

1) S1P is required for vascular health.

2) S1P signaling maintains BBB integrity.

3) apoM transports S1P to diverse vascular beds to enhance signaling.

4) S1P1 signaling at the BBB endothelium is mediated at least in part by S1P1.

The advance presented in this paper is that apoM-associated S1P is also the primary signaling moiety for the BBB. Evidence favoring a role for S1P1 is provided indirectly by acute systemic (i.p.) administration of a selective agonist at S1P1 but not by genetic manipulation of endothelial cell S1P1. Notably the role of S1P1 at the BBB has also been published more than once (point #4 above).

The data are solid. However, the new information is incremental and will primarily be of interests to specialists in this field. The experimental design is quite limited, showing only that ApoM-KO mice show increased paracellular permeability to small-molecule tracers without evident junctional complex disruption, although endothelial thickness is increased. Both pial and penetrating arterioles demonstrate increased transcytosis of labeled BSA. All abnormalities are rescued by acute systemic administration of a small molecule compound selective S1P1 agonist except for pial arteriole transcytosis of BSA.

Essential revisions:

1) In apoM mice BBB transcytosis of albumin (and other plasma proteins) with parenchymal exposure is reported. Given that plasma proteins in brain parenchyma are potently stimulatory and should induce chronic neuroinflammation or tachyphylaxis what is the status of glial elements (microglia, astrocytes, OPCs)?

2) What is the rationale for S1P1 agonist rescuing phenotype of penetrating but not pial arterioles?

3) One conclusion of the study is that manipulating apoM/S1P might show therapeutic benefit. In that regard, how can one interpret the finding that S1P1 agonist rescues the apoM-deficient phenotype without need for apoM?

---

## [Author Response]

Summary:As you will see from the original text of the reviews below, there were substantial differences in the reviewers' judgment on the potential impact of this work. We discussed this topic among the reviewers. There was consensus that the work was of high quality using a sensitive technique, but the following issues must be addressed in the revision along with a point-by-point response to ALL reviewers' comments. As you will see some of these points will likely require additional data or analysis.1) There were places where better quantification is necessary (see point 1 of reviewer 1).

We have now improved our quantification method (see: reviewer 1 comment #4).

2) A more rigorous discussion of the limitations of the study and how it advanced what is already known is needed (see point 2 of reviewer 1).

We have now better related the study to the current knowledge on the topic; underlined new findings throughout the Discussion section, performed additional internal controls (see: e.g. reviewer 1 comment #7); and included a new paragraph on methods limitations (see: Authors comment#1 at the end of the document).

3) A discussion of alternative hypotheses, e.g. other effects of germline apoM deletions or compensation by other HDL components should be included.

This has been now discussed and added to the manuscript (see: reviewer 2 comment #3).

4) It is important to demonstrate whether there was a glial/microglia reaction following entry of plasma proteins (fibrin stain) into the brain parenchyma. (see point of reviewer 3)

We have performed a series of immunostaining analyses that address fibrin content in brain parenchyma and gliosis (see: reviewer 3 comment #1). In addition, as requested, we have identified the parenchymal cells that sequestered albumin in Apom^-/-^ mice (see:reviewer 1 comment #5).

5) Do the authors have data on EM analysis of arterioles and arteries (see point 3 of reviewer 1)?

Unfortunately, quantitative TEM analysis of large vessels is problematic and has not been performed (explained in reviewer 1 comment #3).

Reviewer #1:Sphingosine-1-phosphate (S1P) is a bioactive sphingolipid present in blood, where it is typically bound to either circulating apolipoprotein M (apoM; about 70%) or albumin (about 30%). S1P will preferentially activate different S1P G-protein coupled receptors depending on its binding partner in blood, and apoM-bound S1P preferentially binds to cell surface S1PR1 receptors. Here the authors studied the effect of apoM deficiency on blood brain barrier (BBB) permeability using in vivo two-photon microscopy of extravasated tracers and EM analysis of endothelial tight junctions. Knockout of apoM (apoM-/-) in mice induced increased extravasation of small dye molecules, which could be ameliorated by treatment with S1PR1 agonist SEW2871. EM imaging indicated no changes in endothelial junction length or width in capillaries, but in vivo two-photon microscopy revealed increased numbers of extra-vascular "vesicles" along the arterioles and arteries in the brains of apoM-/- mice, which could be reduced only in the penetrating arterioles with application of S1PR1 agonist SEW2871.This manuscript presents interesting observations in an understudied subject. The manuscript would be improved by addressing the following:Essential revisions:1) The experiments in Figure 1 show increased leakage of the small molecule NaFluo in apoM-/- mice (without quantification). The conclusion in the text indicating that this effect is due to paracellular permeability (subsection “ApoM / S1PR1 signaling deficit increases transcellular transport”) is premature. At this point, all that has been shown is a descriptive illustration of enhanced leakage, not "paracellular permeability."

The purpose of Figure 1 was to introduce to the experimental set-up and data collection process and to demonstrate the time-course of fluorescence accumulation in the brain. Quantifications were shown in Figure 2, as it contained the data from all genotypes and all tested fluorophores. With regard to the properties of NaFluo and paracellular permeability:

NaFluo biochemical properties favor paracellular route over transcellular diffusion or vesicular transport of the molecule across the BBB because:

a) NaFluo is a small (0.365 kDa) hydrophilic molecule, which effectively impedes its transcellular diffusion via the hydrophobic compartment of the lipid endothelial cell membrane.

b) NaFluo is negatively charged, and the electrostatic forces repel the molecule from the negatively charged glycocalyx layer and endothelium cell membrane.

c) In case a small fraction of NaFluo permeates endothelial cell membrane, NaFluo is a strong target for endothelium efflux pumps, i.e. multidrug resistance proteins (MRP) that sequester the tracer back to the blood stream (Hawkins et al., 2007).

Therefore, NaFluo is considered to be suitable for characterizing the BBB paracellular route, independently from transcellular diffusion and vesicular transport across the barrier (Cheng et al., 2010). NaFluo and has been previously used in comparative studies of paracellular and transcellular permeability, e.g. by injections of NaFluo (paracellular route assessment) with albumin-conjugated dyes (transcellular transport assessment) (Hawkins and Egleton, 2006; Kozler and Pokorny, 2003). In addition, our recent work shows that NaFluo (as well as Alexa Fluor 488 and FITC-dextrans) traverse the BBB in vivo with diffusion coefficients and fluorescence distribution that cannot be explained by vesicular transport, but is consistent with mechanisms of passive diffusion of dyes along the concentration gradient across the BBB (Kutuzov et al., 2018).

Also, from subsequent data presented in the manuscript, it would seem that transcellular transport would play a role in increasing NaFluo into the brain.

Although we cannot exclude that some NaFluo molecules can be entrapped in transport vesicle during its formation at the luminal side of the endothelium, in our experiments we did not detect any vesicular fraction of NaFluo, Alexa Fluor 488 and FITC-dx at the BBB. In contrast, albumin-conjugated Alexa488 showed a clear vesicular pattern, which is consistent with transcytosis (De Bock et al., 2016; Minshall et al., 2002; Minshall et al., 2000). Therefore, we consider that the incidental transcytosis of NaFluo, Alexa Fluor 488 or FITC-dx would have a negligible contribution to fluorescence increase, and the effect observed in the brain parenchyma occurs due to the increase in paracellular permeability (See also reviewer 1 comment #7).

This portion of the manuscript (subsection “ApoM / S1PR1 signaling deficit increases transcellular transport”) should be revised;

We have now added the relevant comments above to the Materials and methods and to the Discussion section.

quantified;

We improved our analysis method by accounting for baseline changes and systematic error during sequential injections of fluorophores (see reviewer 1 comment #4).

and/or possibly combined with the data in Figure 2.

We have now combined the data from Figure 1 and Figure 2 into a single figure.

2) These findings seem to be consistent with previous reports showing that activation of S1PR1 reduces endothelial permeability through rearrangement of cellular cytoskeleton and that S1P leads to barrier enhancement via S1PR1 (see for example McVerry and Garcia, 2004; Burridge and Wennerberger, 2004; Singleton et al., 2005).

We did not perform imaging of cytoskeleton in our experiments therefore we did not speculate on *Apom* knock-out effects on cytoskeleton dynamics and BBB permeability. We agree, however, that the cytoskeleton dynamics plays an important role in modulating endothelial barriers, and have now included the following in the Discussion section:

“S1P has been previously shown as an endothelial barrier-enhancing molecule in the lungs (Dudek and Garcia, 2001; Garcia et al., 2001; Schaphorst et al., 2003) and recently, in the brain (Swendeman et al., 2017; Yanagida et al., 2017). Our data ties apoM shortage-induced increase in BBB permeability with S1PR1. The mechanisms of S1PR1-mediated protective effect are not yet fully understood, but may be linked with Rho superfamily of small GTPases. Stimulation of S1PR1 and S1PR2 activates preferentially Rac and Rho, respectively, whilst S1PR3 stimulation activates both GTPases (Windh et al., 1999; Xiong and Hla, 2014). Rac and Rho modulate cytoskeleton dynamics in various cell types (Burridge and Wennerberg, 2004), including the formation and maintenance of a cortical actin ring that stabilizes intercellular junctions between adjoining endothelial cells (Garcia et al., 2001; McVerry and Garcia, 2004). At physiological concentrations, S1P activates preferentially Rac via S1PR1 over Rho (Garcia et al., 2001; Hla, 2003; Shikata et al., 2003), which causes a redistribution of proteins that facilitate cortical actin ring assembly, and consequently tightening of the barrier (Garcia et al., 2001; Mehta et al., 2005). Inhibition of Rac downstream of S1PR1 leads to actin depolymerization, formation of F-actin stress fibers and increased permeability (Singleton et al., 2005; Wojciak-Stothard et al., 2001), and S1PR2 causes Rho activation that inhibits Rac (Dudek and Garcia, 2001; Hla, 2003). However, the formation and disassembly of actin cortical ring affecting BBB permeability should not be simply viewed as Rac vs. Rho antagonism, as inhibition of Rho, similarly to Rac, can reportedly also increase barrier permeability, although without interrupting the actin ring assembly (Etienne-Manneville and Hall, 2002; Garcia et al., 2001). Our quantitative TEM assessment shows that apoM deficiency did not induce changes in junctional complexes ultrastructure. This suggests that the increase of the BBB permeability in Apom^-/-^ mice was either uncoupled from cortical actin ring remodeling, or that during S1PR1 hypostimulation and ongoing S1PR2-mediated Rho inhibition of Rac, Rac activity was still sufficient to maintain the elements of scaffolding complex of cortical actin ring that stabilize junctional complexes.”

Additionally, activated protein C enhances endothelial barriers via EPCR-dependent PAR1-mediated activation of Sphk-1 which generates sphingosine-1-phosphate S1P acting on S1PR1 (Feistritze and Riewald, 2005; Finigan, 2005). Please discuss how the authors observations fit with this previous literature.

We have now included the following in the Discussion section:

“Our in vivo results characterize apoM as a blood-borne molecule that can affect the BBB permeability via S1PR1. In inflammatory conditions, elevated concentrations of blood-circulating thrombin activate protease-activated receptor-1 (PAR1), and subsequently barrier-disruptive metalloproteinases (Bogatcheva et al., 2002; Rempe et al., 2016). On the contrary, low thrombin concentrations activate protein C (APC), a substrate for endothelial cell protein C receptor (EPCR), which alters PAR1 signaling to promote the synthesis of S1P via Sphk-1, and facilitates barrier enhancement (Feistritzer and Riewald, 2005; Riewald et al., 2002). In addition, APC has been shown to also induce S1PR1 phosphorylation, which further promotes the S1PR1 signaling branch in the endothelium (Lee et al., 2001). In line with our results in Apom^-/-^ mice, the barrier enhancement by blood-borne thrombin occurred via S1PR1, as silencing of S1PR1 but not e.g. S1PR3 abolished the APC-mediated effect (Feistritzer and Riewald, 2005; Finigan et al., 2005). Except for apoM and thrombin, S1P signaling can be modulated by hyaluronic acid (HA). Depending on the form, i.e. high-molecular weight-HA (HMW-HA) or low-molecular weight (LMW-HA), the complex promotes interaction of a cell-surface glycoprotein CD44 isoforms with S1PR1 or S1PR1/3, respectively (Singleton et al., 2006). This leads to enhancement of the barrier function via S1PR1 or an increase in BBB permeability via S1PR3 due to transactivation (i.e. phosphorylation) of respective receptors by CD44 isoforms, independently of S1P levels (Singleton et al., 2006). This further corroborates our findings on the regulation of S1P signaling by blood circulating molecules via preferential activation of specific S1P receptors.

However, the analogies should be made with caution, as the mechanistic insights come primarily from in vitro barrier models or non-brain tissue, where endothelial barrier properties may differ from the BBB in vivo. For example, Rac and Rho are known to elicit cell-type and subcellular localization-specific effects (Burridge and Wennerberg, 2004; Donati and Bruni, 2006; McVerry and Garcia, 2004), and their intracellular pathways are cross-linked with numerous regulatory feedback loops, including e.g. Rho inhibiting Rac and vice versa (Burridge and Wennerberg, 2004; Sander et al., 1999). This adds to the complexity of interpretation of Apom knock-out effects on the BBB and is currently an active area in research.”

*3) The EM analysis of endothelial junctions is very nicely done, but performed only at the capillary level. Since the authors found that arterioles and arteries show the most pronounced changes in vesicular transport between genotypes with* in vivo *imaging, should these vessels be also checked for endothelial junction integrity in apoM-/- mice?*

We have aimed to characterize endothelial junction ultrastructure in large vessels (ø 25-120 µm) together with capillaries. However, in contrast to capillaries, large vessels were susceptible to perfusion-fixation artifacts (non-uniform distortion of endothelium neighboring large tissue-devoid areas, i.e. vessel lumen), which rendered pial and penetrating vessels not suitable for TEM quantitative assessment of length, thickness, coverage of tight junctions and the area of the endothelium. This information has been now added to the Results section.

4) In the experiments presented in Figure 2, three tracers with similar fluorescence spectra are sequentially injected into the vasculature. Injection of one tracer after another of the same color has the potential to confound data of the following tracer injections. Have the authors checked that previous tracers injected have cleared before injecting the next tracer in these experiments? Is a baseline taken before each subsequent tracer injection?

The time of bloodstream clearance differs between fluorophores and varies between animals. Waiting for tracer clearance would introduce a large systematic error where the time of clearance of a preceding dye affects the timepoint of imaging of a consecutive dye(s). Consequently, this could obscure characterization of the biological effects of SEW2871 on the BBB, as the assessment of permeability in each mouse for each respective tracer would be performed at different time after the drug administration. SEW2871 has a relatively short half-time in the blood plasma with its biological effects peaking at 1 hour, and returning to baseline 4.5 hours after administration (as determined by e.g. Akt phosphorylation)(Jo et al., 2005). Instead, we chose to standardize the time of imaging, i.e. FITC-dx, Alexa Fluor 488 and NaFluo were always imaged 2.5 hours, 3 hours, 3.5 hours after SEW2871 administration, respectively. While this approach is also susceptible to a systematic error that can originate from consecutively injected dyes accumulation, we have now revised our analysis to account for this possibility.

In our revised model, we estimate the contribution of a preceding fluorescence accumulation to subsequent fluorophore(s) signal measurement. We have previously assumed that the fluorescence in the brain parenchyma of a preceding fluorophore(s) did not have a significant effect on the signal on subsequent fluorophore(s). This was based on a preliminary assessment of the data, where the increases of preceding fluorophores were within the standard error of mean changes of a subsequent fluorophore (Author response image 1).

**Author response image 1. respfig1:** Raw fluorescence signal collected from brain parenchyma after sequential injections of FITC-dx, Alexa Fluor 488 and NaFluo. A.U. = arbitrary fluorescence unit (detector counts). The time between data points of last frame in Z-stack of preceding fluorophore and time of last Z-stack of subsequent fluorophore was 2 min (1 min of injection + 1 min of Z-stack collection time). n=7 animals in each experimental.

Now, for each mouse, we subtracted the constant autofluorescence background from FITC-dx, Alexa Fluor 488 and NaFluo traces and performed an exponential fit to the FITC-dx trace that was extrapolated to subsequently imaged Alexa Fluor 488 and NaFluo (Author response image 2). We have deliberately chosen the exponential fit to account for the largest systematic error in our experiment, i.e. the largest theoretical contribution of a preceding tracer to a subsequent tracer signal. The fluorescence is additive, therefore we corrected Alexa Fluor 488 and NaFluo traces by subtracting FITC-dx extrapolated fit values (Author response image 2) and effectively, uncoupled FITC-dx kinetics from Alexa Fluor 488 and NaFluo. Next, to the corrected trace of Alexa Fluor 488 we performed an exponential fit and extrapolated it to the time of NaFluo imaging. Similarly, as before, we subtracted Alexa Fluor 488 extrapolated fit from NaFluo trace (Author response image 2).

**Author response image 2. respfig2:** Correcting for the preceding fluorophore signal contribution to the subsequent fluorophore signal measurement. (**a**) Traces from three consecutive injections of fluorophores with a curve fit to FITC-dx trace. The fit is extrapolated and subtracted from Alexa Fluor 488 and NaFluo traces. (**b**) Second fit is performed to a corrected Alexa Fluor 488 trace, extrapolated and subtracted from NaFluo trace. The model is applicable for both, relatively low (blue traces, WT animal) and relatively high (red traces, apoM-/- mouse) fluorescence changes.

The same approach was used to account for baseline fluorescence changes in the bloodstream (Figure 1—figure supplement1C-D).

We have performed area under curve (AUC) analysis on both types of data, i.e. arbitrary fluorescence units [a.u.] (Figure 1—figure supplement1E-F) and relative change in fluorescence [F/F0] (Figure 1). The results were consistent with our previous notion, i.e. there was no significant leakage of FITC-dx, but an increase of BBB permeability for Alexa Fluor 488 and NaFluo tracers in *Apom^-/-^* mice, and that the effect could be reversed by SEW2871 treatment.

We chose to report the data as relative changes for the results to be compatible between different imaging setups. The revised model has been now described in the Materials and methods section with a new supplementary figure (Figure 1—figure supplement 1).

5) In the BSA-Alexa 488 transcellular experiments illustrating vesicles shown in Figure 4, could the authors confirm the cell type taking up the BSA in the brain with histology?

We have performed a series of immunostaining analysis on perfusion-fixated brains of

*Apom^-/-^* mice. We tested for colocalization of cell-internalized BSA-Alexa488 in the brain parenchyma with expression of Iba-1 (microglia/macrophages); TMEM119 (microglia-specific); RM0029-11H3 (macrophage-specific) and GFAP (astrocytes). We have successfully identified the BSA-Alexa488 sequestering cells to be macrophages (Author response image 3), and found only minor signs (single puncta) of BSA-Alexa488 uptake by other cell types.

**Author response image 3. respfig3:** BSA-Alexa 488 uptake by Iba-1 and RM0029-11H3 positive cells, i.e. macrophages.

These results, including additional immunostaining assays, have been added now to the manuscript subsection “ApoM / S1PR1 signaling deficit increases transcellular transport”; (Figure 3—figure supplement 2).

6) Additionally, the BSA-Alexa 488 transcellular data would be strengthened by adding a conformation of the observations using a different method, e.g. immunostaining or EM imaging.

Albumin-based assay is a well-established method to monitor vesicular transcytosis and has been used previously to describe and discern BBB transcellular and paracellular BBB permeability, e.g. in (Knowland et al., 2014). Here, we imaged exclusively the fraction of albumin that circulates in the bloodstream, associates to, and crosses the BBB. Recent evidence suggests that albumin can also be produced in CNS by microglia during neuroinflammatory responses (Ahn et al., 2008). In contrast to blood-borne BSA-Alexa488, it is possible that albumin immunostaining would reveal the total albumin content in the brain, including albumin produced in the parenchyma, thus could potentially obscure the assessment of transcellular BBB permeability.

7) Why were the authors able to see the larger BSA-Alexa 488 tracer cross the BBB, but not the FITC-dextran?

In contrast to FITC-dextrans, transport of albumin (~70kDa) depends on specific interactions of the protein with endothelium. It is dissociated from the diffusion of small molecules via the paracellular route since the space between adjoining endothelial cells is not sufficiently wide to permit albumin leakage. Transport of albumin occurs either by fluid-phase absorption or by selective receptor-mediated transcytosis that involves albumin-docking glycoprotein gp60 and caveolin-1 -mediated formation of albumin-transporting vesicles (De Bock et al., 2016; Mehta and Malik, 2006; Minshall et al., 2002; Minshall et al., 2000; Xiong and Hla, 2014). As discussed in reviewer 1 comment #1b, it is theoretically possible that a fraction of dextran-conjugated fluorophores may be internalized by albumin transport vesicle and cross the BBB, however our data does not support it (Author response image 4).

**Author response image 4. respfig4:** Additional internal control: the graph shows the fluorescence intensity profile plot of n=10 albumin-containing vesicles. There was no enrichment of TRITC-dx signal (red) in BSA-Alexa488 –containing vesicles (green) at the BBB.

8) The finding that the endothelial thickness is increased with apoM knockout is interesting. What might be the reason for this? Swelling or something else? Please discuss.

We have now included the following section to the discussion section:

“The small (~10%) increase in endothelium thickness in Apom^-/-^ mice may be due to the impaired S1PR1 -dependent formation and the maintenance of the cortical actin ring; and formation of actin stress fibers facilitated by S1PR2 activation (Garcia et al., 2001; Mehta et al., 2005; Singleton et al., 2005; Wojciak-Stothard et al., 2001). Indirectly, BBB permeability changes may be due to osmotic imbalance and cell swelling, but it is uncertain whether this occurs in the brains of Apom^-/-^ mice. Lack of S1PR1 activation aggravates edema in acute brain and lung injury (Ito et al., 2019), but here the system may not be challenged enough to induce osmotic swelling, as Apom^-/-^ mice have no signs of lung edema and exhibit normal microvasculature morphology (Christoffersen et al., 2011). Other processes may also influence endothelium thickness, but their relevance in Apom^-/-^ mice is yet to be established. For example, S1PR1 activity counteracts vascular endothelial growth factor (VEGF) -mediated endothelium contraction in angiogenesis (Gaengel et al., 2012), but whether S1PR1 hypostimulation affects VEGF signaling in the adult brain in Apom^-/-^ mice is unknown. In addition, impaired S1PR1 signaling activates neuroinflammatory factors (Blaho et al., 2015; Blaho and Hla, 2014), and e.g. TNF-α has been shown to alter cytoskeleton and hyperplastic properties of endothelial cells (Kang et al., 2008; Stroka and Aranda-Espinoza, 2011). Thus, it cannot be excluded, that the changes in endothelium morphology result from both: direct effect of S1PR1 hypostimulation on endothelial cell, and indirect effects linked with impaired BBB-induced loss of brain homeostasis.”

Reviewer #2:Janiurek et al., studied the BBB function of WT and ApoM KO mice. They show, using two-photon microscopy, that ApoM KO mice exhibit a BBB defect in small molecules. This was expected since we know from studies of Yanagida et al., that S1PR1 ECKO also show this phenotype. The present manuscript provides a beautiful demonstration of this using state of the art, live animal imaging studies. The data are convincing and demonstrate that ApoM / S1P signaling via the endothelial S1PR1 is important in controlling the BBB function of small molecular weight compounds. The authors also examined the endothelial morphology and junctions as well as transcytosis in various vessel subtypes. They report different changes in transcytosis in WT and ApoM KO in penetrating arterioles vs capillaries.Overall, the study is well done and impressive. Some of the results are expected since we already know that S1PR1 regulates the same function in the BBB.

We would like to thank the Referee for the nice comment. We have now better addressed the novelty of the study in the manuscript, underlining, among others, our methodological approach and vessel-specific effects. We would also like to note that while Yanagida and collegues addressed the role of S1PR1 in the maintenance of the BBB, our study is focused on the upstream regulator of the S1PR1, i.e. apoM, which, as we show, exhibits its actions on the BBB via S1PR1. In contrast to S1PR1^ECKO^, *Apom^-/-^* mice have still functioning S1PR1, thus S1PR1 knock-out is not necessary to induce the incease in BBB permeability, as even hypostimulation of the S1PR1 is sufficient to cause the loss of the barrier function, which (to our knowledge) has not been previously shown.

It would be useful if the authors were to extend the discussion they had with S1PR2. For example, does S1PR2 inhibitor inhibit the increased permeability seen in ApoM KO mice? Since they have the techniques to examine this issue, this would add much to the field.

We did not perform experiments that involve S1PR2 inhibition. However, we would like to thank the referee for the suggestion, as modulating S1PR2 may be an excellent point of entry for a follow-up study. We have now added the following text to the manuscript that continues the discussion (Discussion section).

“The hypostimulation of S1PR1 in Apom^-/-^ mice is likely to shift the signaling equilibrium towards cellular pathways that promote activation of Rho (via S1PR2) over Rac (via S1PR1), and the effects of increased BBB permeability are likely to be the net outcome of the failure in homeostatic interplay between receptor-dependent branches of S1P signaling, rather than simply a decrease in Rac activity. Consequently, the observed impairment of the BBB cannot simply be linked to a reduction in the amount of circulating S1P, but more specifically to the lack of apoM-bound S1P in favor of albumin-bound S1P in the blood. This notion is further supported by the result of the selective stimulation of S1PR1, which restored the BBB permeability in Apom^-/-^ mice to the level of healthy controls. Given that inhibition of S1PR2 tightens the endothelial barrier in lungs by facilitating cortical actin ring assembly and stabilization of tight junctions (Sanchez et al., 2007), and in the brain supports the BBB functional integrity (Cao et al., 2019; Kim et al., 2015), it is plausible that S1PR2 stimulation may alleviate BBB phenotype in Apom^-/-^ mice. It should be noted, however, that S1PR1 expression in the endothelium is higher in the arterial branch of cerebral vasculature compared to venules, whereas S1PR2 expression is higher in venules than in arterioles (Vanlandewijck et al., 2018). It may lead, similar as in SEW2871 treatment, to region-specific effects, where the BBB responses to S1PR2 modulation may differ between distinct types of cerebral microvessels.”

There are some errors, such as G proteins downstream of S1PR1 and S1PR2 and not GPCRs.

We thank for pointing it out and we believe we have corrected now all the minor errors.

Also, a better discussion of HDL/ ApoM function and BBB would be of interest.

We have now added the following section to the manuscript discussion section.

“High-density lipoprotein (HDL) particles are heterogeneous in size and content of apolipoproteins and lipids. In humans, HDL plasma concentrations correlate with reduced susceptibility to cerebro- and cardiovascular diseases (Fellows et al., 2015; Hovingh et al., 2015; Rader and Tall, 2012), and approximately 5% of HDL contains apoM (Christoffersen et al., 2006; Xu and Dahlback, 1999). The beneficial effects of HDL can be, in part, explained by HDL association to apoM/S1P. We show that lack of apoM increases the flux of small molecules across the BBB, similar to previous findings investigating the permeability in lung and adipose tissue (Christensen et al., 2016; Christoffersen et al., 2018; Christoffersen et al., 2011). Stressing the system even further, lack of apoM increases the susceptibility to neuroinflammation (Blaho et al., 2015; Galvani et al., 2015). Mainly this has been interpreted as effects caused by blood-circulating apoM/S1P complex. However, a recent study suggested that porcine brain endothelial cells express and secrete apoM into the brain parenchyma (Kober et al., 2017). This has not been reported in other species but could also play a role in the increased permeability observed in the present study. Except S1P, apoM can potentially bind other ligands, e.g. retinoic acid (Ahnstrom et al., 2007) or oxidized phospholipids (Elsoe et al., 2012). Lacking apoM would increase free oxidized phospholipid content, which potentially can also affect the BBB (Singleton et al., 2009). However, reversing the BBB phenotype by SEW2871 suggests that the observed effects are primarily mediated via S1PR1 rather than e.g. oxidized phospholipids. Noteworthy, with the absence of apoM, S1P may still associate to HDL. A recent study showed that in mice lacking expression of both, apoM and albumin, S1P binds to HDL via novel S1P binding protein apoA4 (Obinata et al., 2019). This may potentially constitute a compensatory mechanism in Apom^-/-^ mice, but whether HDL/apoA4/S1P complex also plays a role in maintaining the BBB is unknown and will need further investigations.”

Reviewer #3:Essential revisions:1) In apoM mice BBB transcytosis of albumin (and other plasma proteins) with parenchymal exposure is reported. Given that plasma proteins in brain parenchyma are potently stimulatory and should induce chronic neuroinflammation or tachyphylaxis what is the status of glial elements (microglia, astrocytes, OPCs)?

*S1pr1* conditional KO mice exhibit no signs of reactive astrogliosis (Yanagida et al., 2017), despite the increase in pro-inflammatory markers (Galvani et al., 2015). In contrast to *S1pr1* conditional KO, in *Apom^-/-^*mice the S1PR1 signaling pathway is not abolished. On the other hand, *Apom^-/-^*mice have impaired S1PR1 signaling during embryonic development, unlikely to *S1pr1* conditional KO. We have now assessed the fibrinogen content in the brain parenchyma; and morphological features of microglia/macrophages (Iba-1) and astrocytes (GFAP).

We detected no signs of fibrinogen entry into the brain parenchyma; and similarly to *S1pr1* conditional KO (Yanagida et al., 2017), the morphology of microglia/macrophages and astrocytes was similar between WT and *Apom^-/-^*mice. This data suggests a lack of gliosis and has been now added to the manuscript subsection “ApoM / S1PR1 signaling deficit increases transcellular transport”; Figure 3—figure supplement 1).

2) What is the rationale for S1P1 agonist rescuing phenotype of penetrating but not pial arterioles?

This section has been now added to Discussion section.

“The susceptibility of the arterial branch of the microvasculature to apoM shortage may be linked to preferential expression of S1PR1 and APC in pial and penetrating arterioles (Vanlandewijck et al., 2018), thus possibly high reliance in these vessel types on S1PR1 signaling in the maintenance of the BBB. It is unclear, however, why penetrating, but not pial arterioles responded to SEW2871, but anatomical differences may be of importance. For example, S1P signaling promotes the formation of the glycocalyx, which limits the diffusion of hydrophilic and negatively charged substances, and supports the barrier function (Zhang et al., 2016). Glycocalyx in pial arterioles is estimated to be thicker compared to the penetrating arterioles (Yoon et al., 2017), but it is unknown how S1PR1 hypostimulation affects distribution and coverage of glycocalyx layer in Apom^-/-^ mice. In addition, the reconstitution of barrier function may have a different time course in pial and penetrating arterioles. Lastly, in contrast to bolus SEW2871 injection, a sustained stimulation of S1PR1 may be needed to restore BBB function in pial arterioles during ongoing apoM deficit.”

3) One conclusion of the study is that manipulating apoM/S1P might show therapeutic benefit. In that regard, how can one interpret the finding that S1P1 agonist rescues the apoM-deficient phenotype without need for apoM?

We used the term “phenotype” to describe the loss of BBB functional integrity (as in “BBB phenotype”), not all systemic effects that may be caused by apoM shortage. Regarding SEW2871, the therapeutic effect with the absence of apoM can be explained by the fact that in contrast to S1P, SEW2871 does not require apoM as a carrier in the bloodstream, and does not require apoM to activate S1PR1.